# A Feasibility Study on the Use of Injection Molding Systems for Mass Production of 100W Class Wind Turbine Blades

Joongseon Kim [1], Upendra Tuladhar [1], Deokkeun Je [2], Marinus Mieremet [3], Joonho Baek [3], Hoseong Ji [4,*] and Seokyoung Ahn [1,*]

1   School of Mechanical Engineering, Pusan National University, 2 Busandaehak-ro 63 beon-gil, Geumjeong-gu, Busan 46241, Republic of Korea; kjs16852002@naver.com (J.K.)
2   Jesan Precision Mold, 41 Sinho Industrial Complex 3-ro 88beon-gil, Gangseo-gu, Busan 46759, Republic of Korea
3   Department of Engineering Research, ESCO RTS, Seoul 06134, Republic of Korea
4   Nuclear D&D Core Research Center, Pusan National University, 2 Busandaehak-ro 63 beon-gil, Geumjeong-gu, Busan 46241, Republic of Korea
*   Correspondence: hsji@pusan.ac.kr (H.J.); sahn@pusan.ac.kr (S.A.); Tel.: +82-51-510-2494 (H.J.); +82-51-510-2471 (S.A.)

**Abstract:** A feasibility study on the mass production of a small wind turbine blade using an injection molding process was conducted. The blade was divided into three sections suitable for injection molding, and the mold was designed and analyzed using Moldflow CAE S/W. The optimal feedstock material was selected through comparison and analysis of three candidate materials. A mold was manufactured to test the injection molding process and evaluate related parameters. The resulting blade was assembled with other components, and a generator was installed to assess its durability, safety, and performance under various conditions. The results indicated the feasibility of producing a blade for a small wind turbine through injection molding, which predicted higher productivity and lower costs compared to traditional manufacturing methods that rely heavily on manual labor.

**Keywords:** small wind turbine; mass production; injection molding; Moldflow



## 1. Introduction

The increasing global demand for energy due to economic and technological advancements has led to the widespread use of fossil fuels, which not only face depletion but also have serious adverse effects on the environment through the emission of pollutants [1–3]. To mitigate these impacts, there has been a shift towards the exploration of renewable and clean energy sources such as solar, wind, and solar-hydrogen energy [4]. The Intergovernmental Panel on Climate Change (IPCC) has recognized the need for action and has proposed regulations to reduce carbon emissions from fossil fuel combustion [5]. This has resulted in increased awareness and effort to achieve carbon neutrality and perform research on sustainable renewable energy sources.

Wind power is a widely used clean and sustainable energy source that is economically and environmentally favorable. It generates energy through the harnessing of natural wind and has a lower impact on natural ecosystems compared to other energy sources [6,7]. However, wind turbines have been known to cause harm to wild animals through collisions and also generate noise and visual interference in the surrounding environment [7,8]. For wind turbines to operate optimally, they require an open area with no turbulence, wind, or damage risk [9]. Small wind generators, on the other hand, have a smaller installation footprint and produce less noise. Studies are being conducted to explore the potential for eco-friendly urban development through the widespread installation of small wind generators in cities. Bukala et al. found that small wind turbines have high investment value, and Ishugah et al. highlighted the economic and environmental advantages of small wind turbines operating in urban environments [10,11]. This has led to ongoing research into the development of various small wind power generators [12].

In keeping with current global trends, researchers have studied a small wind power generation system in the form of a spiral-shaped turbine that utilizes Archimedes' helical water turbine principle. By conducting a theoretical analysis of the spiral blade and its response to wind angle [13,14], a turbine was developed that adjusts to the optimal position automatically, similar to a wind vane. Initially, the spiral blade was fabricated using fiber-reinforced plastic (FRP) through a hand lay-up method, which posed several issues. The manual nature of FRP manufacturing makes it difficult to achieve consistent quality, and trapped bubbles between the resins can cause cracks after the resin has hardened and been laminated. Furthermore, the time-consuming process of hardening and laminating the resin made it unsuitable for mass production. Despite efforts to improve productivity and quality through vacuum-assisted resin transfer molding (VARTM) [15,16], it was found to not be highly productive. In view of this, injection molding was proposed as an alternative method to enhance both productivity and product quality.

The injection molding process is a highly productive method of mass production that can result in products with consistent quality. It involves the use of a molding machine to inject a flowable plastic resin into a mold to create a part. This process is widely used in a variety of industries and is relevant to many aspects of daily life. It is employed in the production of micro-scale products such as medical supplies and semiconductor packaging, as well as larger-scale products such as automobiles and aircraft parts.

Injection molding process simulation and analysis (Computer Aided Engineering: CAE) have been widely studied to determine the feasibility of injection molding and predict product deformation [17,18]. In addition, in terms of research methodology, studies have been conducted to select appropriate materials for injection molding by inputting the characteristics of various feedstock materials into an analysis program prior to the actual molding process, thus reducing trial-and-error efforts [19,20].

However, there is a concern that using only thermoplastic resin in the injection molding process may result in blades with insufficient strength to withstand wind pressure. To address this issue, researchers needed to research whether a combination of thermoplastic resin and glass fibers during the injection molding process could create a blade with sufficient strength.

The objective of this study was to evaluate the viability of fabricating a prototype of the blades for a spiral-shaped small wind power generator through injection molding. The mold and cooling channels were designed and manufactured, and computational analysis of injection molding was performed using Moldflow. This analysis aimed to determine the feasibility of the injection molding process and identify potential product defects. Furthermore, Moldflow was utilized to compare the mechanical properties of three different materials, and the best-suited feedstock was selected for the molding process. Finally, the functionality of the produced blade was assessed by incorporating it into the spiral small wind generator.

## 2. Small Wind Turbine Design

The small wind turbine system is depicted in Figure 1, with its working mechanism illustrated with its components.

The turbine has a spiral design aimed at spiraling the incoming wind. The blade of the small wind turbine is designed to adjust to the direction of maximum wind flow via a yawing system. Prior to the injection molding process, the shape of the blade was modified to allow for its successful production through injection molding. The materials used in each component of the small wind turbine are shown in the dismantling diagram in Figure 1b.

The blade is attached to the central shaft, and a fixing rod is placed in its center to prevent wind-related damage. To optimize the injection moldability, the blade was divided into three parts which were then assembled after molding. The mold design for the divided blade is presented in Figure 2a,b. The total size of the mold was $750 \times 990 \times 1135$ mm$^3$, and the fixed and movable sides were separated to facilitate blade ejection during the molding process.

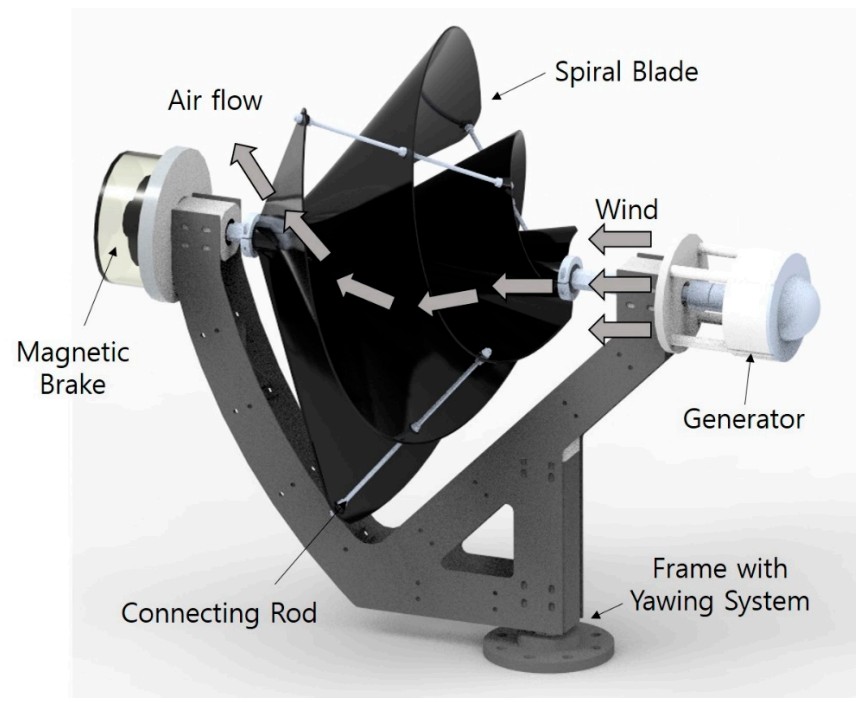

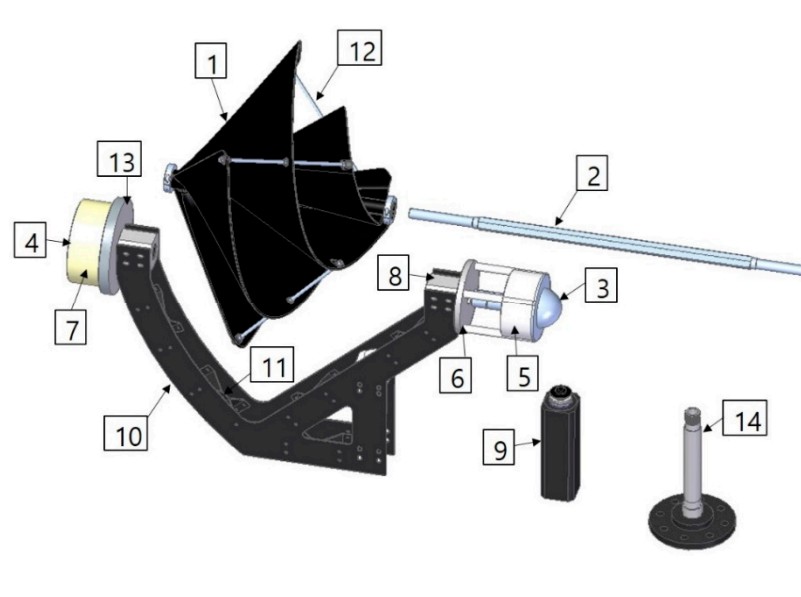

| | Part | Material |
|---|---|---|
| 1 | Blade | Injection Molding |
| 2 | Shaft | SM45C |
| 3 | Generator case | Arcyl |
| 4 | Brake case | Arcyl |
| 5 | Generator | SUS304 |
| 6 | Generator plate | Aluminium 6061 |
| 7 | Brake | SM45C |
| 8 | Bearing housing | Aluminium 6061 |
| 9 | Yawing deck case | SS400 |
| 10 | Frame | SUS304 |
| 11 | Connecting part | SUS304 |
| 12 | Fixing rod | SUS304 |
| 13 | Brake plate | Aluminium 6061 |
| 14 | Yawing shaft | SS400 |

**(b)**

**Figure 1.** Small wind turbine 3D design configuration and the yawning system with its components. (**a**) Small wind turbine 3D design configuration; (**b**) Airflow self-aligning via the yawing system.

For a split turbine blade, designing a dividing line along its curved shape is crucial. Horizontally dividing the mold can pose difficulty in removing the product, so the mold was split into individual layers and divided along the blade edge. The shaft part, which is assembled to the shaft, was placed in the center of the movable mold. An ejector pin was added to the movable side to prevent cracking during the ejection stage.

Although the product's geometric shape and single-layer mold, and central assembly shape were reflected in the mold design, an evaluation was needed to determine if it was actually possible to inject. In this study, Moldflow CAE analysis was used to investigate the feasibility of moldability.

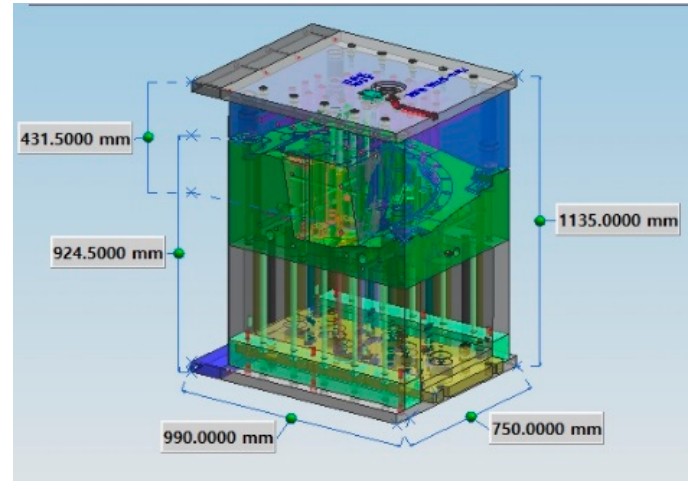

(**a**)

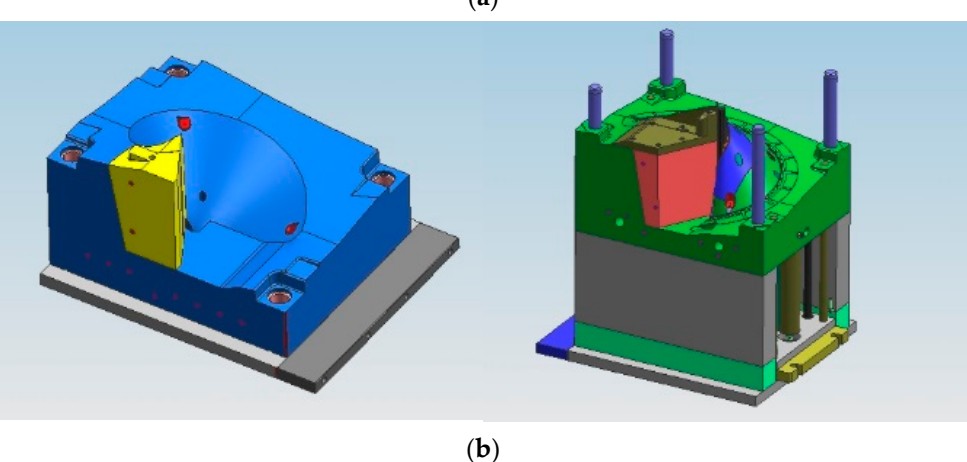

(**b**)

**Figure 2.** Injection mold 3D design apparatus. (**a**) Small wind turbine 3D design configuration; (**b**) Airflow self-aligning via the yawing system.

## 3. CAE Analysis

The feasibility of injection molding the divided spiral blade and its associated product-related issues were assessed using Moldflow Adviser 2018. Prior to the analysis, the product configuration was 3D modeled, the cooling system was incorporated, and the model underwent preprocessing.

### 3.1. Turbine Blade and Cooling System Design

Based on the injection mold cavity and cooling system information, the design was performed as shown in Figures 3 and 4 for a 750 mm split wind turbine. The gate location was selected by the mold engineer's empirical know-how.

The cooling system, designed to match the blade shape, is displayed in Figure 4. The dimensions of the split wind turbine are 558.5 mm along the X-axis, 385.62 mm along the Y-axis, and 476.494 mm along the Z-axis. Additionally, the center axis is divided into three spirals along the curved blade's contact portion. To account for the shrinkage of the spiral blade after injection molding, the cooling system was designed close to the surface of both the inner and outer blades to allow for maximum cooling of the spiral blade within the mold.

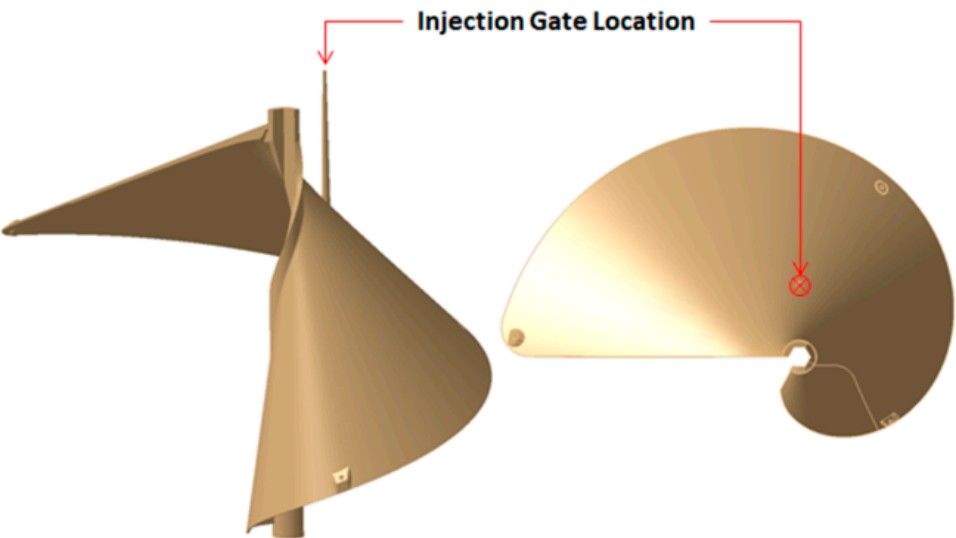

**Figure 3.** Split wind turbine blade 3D model and the location of the injection gate.

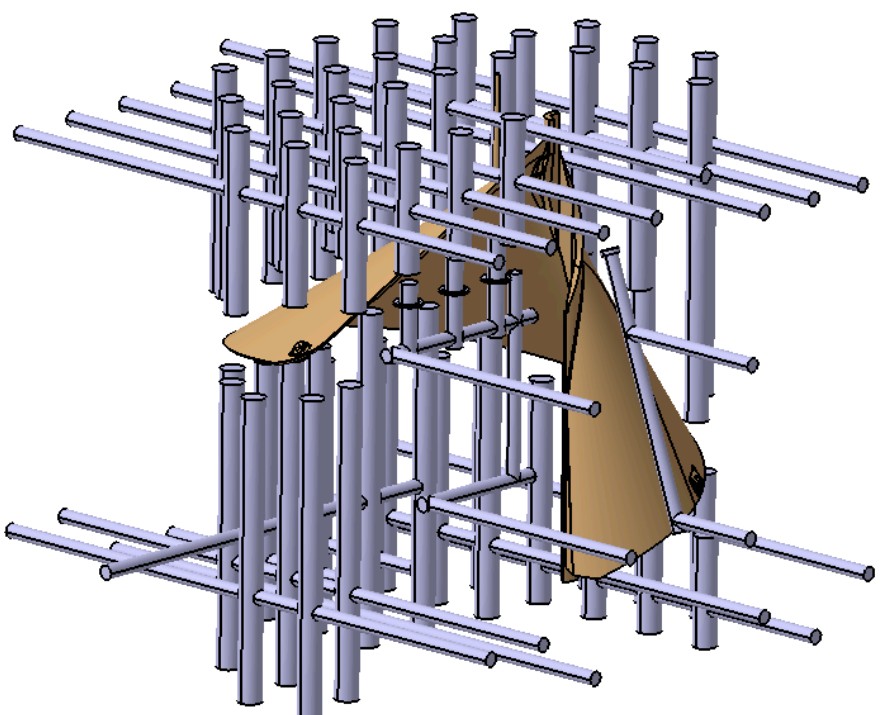

**Figure 4.** Cooling system design for the mold.

### 3.2. Preprocessing of the 3D Model and Analysis Setup

The 3D model preprocessing for mesh generation and analysis setup was carried out for refrigerant configuration, inputting gate information, and specifying the injection materials.

The mesh generation results are presented in Table 1, which were generated by using the Match and Dual domain options of Moldflow. The mesh generation data confirms the successful generation of the mesh without any issues regarding mesh quality.

**Table 1.** Mesh generation result summary.

| Items | | Numerical Values |
|---|---|---|
| Entity triangles | | 59,020 |
| Connected nodes | | 29,514 |
| Connectivity regions | | 2 |
| Invisible triangles | | 0 |
| Aspect ratio | Maximum | 49.13 |
| | Average | 1.76 |
| | Minimum | 1.16 |
| Surface area | | 4705.85 cm$^2$ |
| Volume | | 1687.3 cm$^3$ |

For the feedstock, the following three materials were selected,

- Supran 1340, SAMBARK LFT Co., Ltd. (Republic of Korea)
- RB84HP, Hanwha Total Petrochemical Co., Ltd. (Republic of Korea)
- TEKAFINTM P730G45, Mando Advanced Materials (Republic of Korea)

The analysis was conducted using Themylene P7−45FG−0791 BK712 as a substitute for TEKAFINTM P730G45, which was not present in the Moldflow library. P7−45FG−0791, composed of propylene (PP) and 45% glass fiber, has comparable physical and mechanical properties to TEKAFINTM P730G45. The mold temperature and melt temperature for P7−45FG−0791, at 55 °C and 230 °C respectively, align with those of TEKAFINTM P730G45, and its elastic modulus of 8202 MPa is also similar to that of TEKAFINTM P730G45. The shrinkage of P7−45FG−0791 is greater than that of TEKAFINTM P730G45, making it safer to predict shrinkage and deformation.

Figure 5 shows The visibility-shear rate graph and 2-states Tiet model pVT curve of Supran 1340 material. Supran 1340 material demonstrates viscous thinning behavior and is identified as a crystalline resin. Similarly, RB84HP and P7−45FG−0791 also exhibit similar trends.

Table 2 presents the physical properties, while Table 3 displays the mechanical properties of each material. All three materials are based on PP with the addition of 40% glass fiber in Supran 1340 and 45% glass fiber in P7−45FG−0791. The singular gate is located as depicted in Figure 3 and has a cold/circular tapered configuration. The processing parameters for injection molding are separated according to the existence of glass fibers, as presented in Table 4.

**Table 2.** Material physical properties.

| Material ID | Supran 1340 | RB84HP | TEKAFINTM P730G45 | Themylene P7−45FG−0791 BK712 |
|---|---|---|---|---|
| Alternative material (manufacture) | - | - | Themylene P7−45FG−0791 BK712/ (Asahi Kasei Plastics NA Inc.) | - |
| Family abbreviation | | | PP | |
| Material structure | | | Crystalline | |
| Fibers/fillers | 40% glass fiber | - | 45% glass fiber | 45% glass fiber |

**Table 3.** Mechanical properties of the feedstock.

| Material ID | Supran 1340 | RB84HP | TEKAFINTM P730G45 | P7−45FG−0791 |
|---|---|---|---|---|
| Elastic modulus (MPa) | 9726.03 | 1340 | 8728 | 8202 |
| Shear modulus (MPa) | 2018.68 | 481.3 | - | 1847 |
| Shrinkage (%) | 0.4–0.9 | 0.58–0.99 | 0.3–0.8 | 0.78–1.44 |
| Melt temperature (°C) | 245.0 | 210 | 230 | 230 |

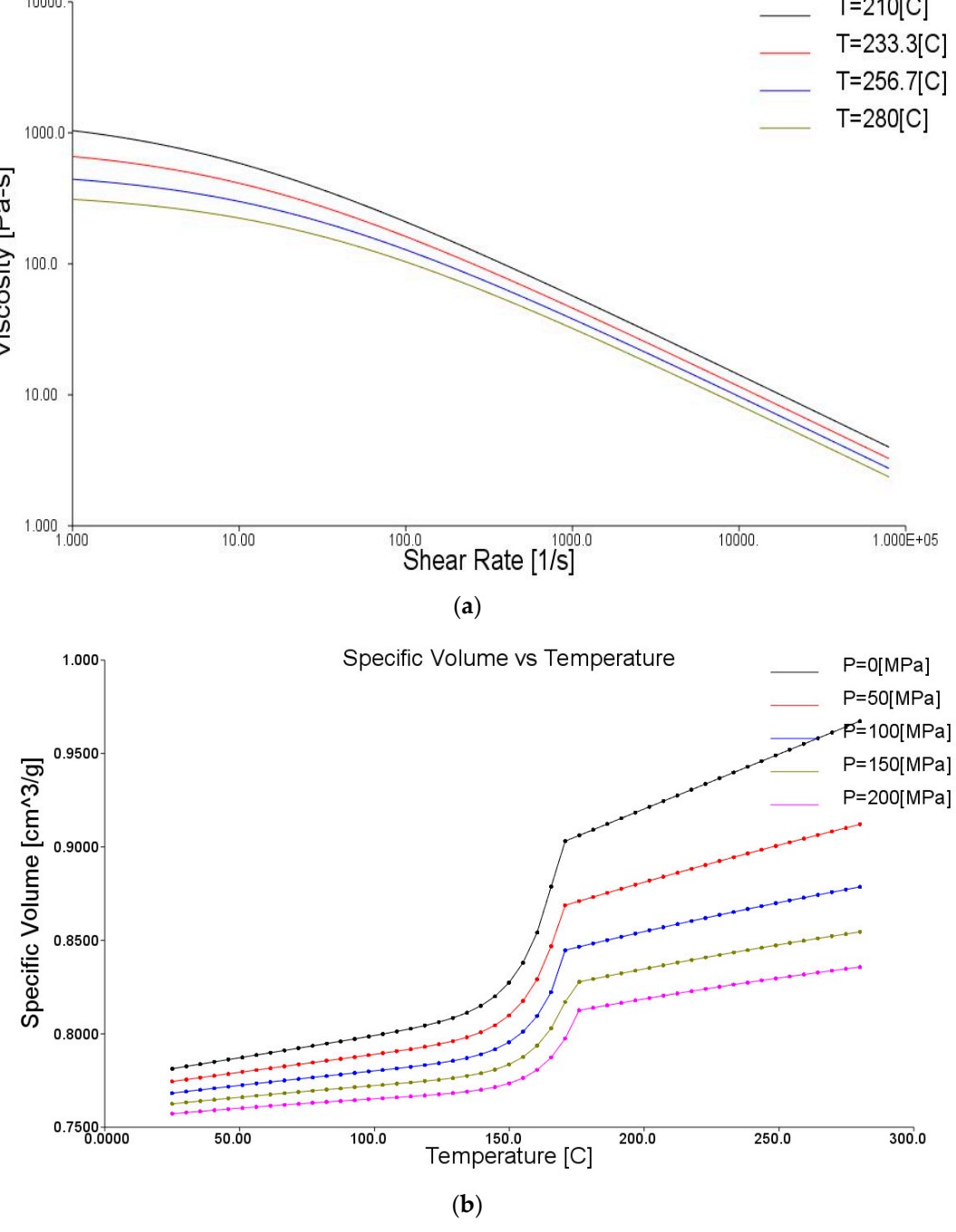

**Figure 5.** Viscosity-shear rate graph and 2-states Tiet model pVT curve of Supran 1340 material. (**a**) Viscosity-shear rate with varying temperatures; (**b**) 2-states Tiet model pVT curve with varying pressures.

**Table 4.** Summary of the processing parameters.

| Material ID | Supran 1340 and P7−45FG−0791 | RB84HP |
|---|---|---|
| Mold temperature (°C) | 55.0 | 45.0 |
| Max. machine injection pressure (MPa) | 180.000 | 180.000 |
| Injection time selected | Automatic | Automatic |
| Velocity/pressure switch-over | Automatic | Automatic |
| Ejection temperature (°C) | 115 | 121 |

## 4. Analysis Results and Material Selection

A Moldflow analysis was performed using the input processing parameters and input information to evaluate the results. This simulation was chosen for the Fill + Pack + Cool + Warp analysis. The most appropriate material for the spiral blade was chosen based on the analyzed data and the mechanical properties of the injection material.

### 4.1. Cooling Channel Performance

One of the most important factors in injection molding is the cooling channel. The temperature of the mold varies due to the design and conditions of the cooling channel, which has a great influence on injection molding. It affects many factors, such as fill time, short shot, ejection time, and deformation of the product; various studies are being conducted on cooling channels [21–24]. In this study, cooling channel performance analysis was also conducted to evaluate the performance of the cooling channel.

An analysis of the cooling channel performance was carried out using water as a coolant with a flow rate of 10 L/min and a temperature of 25 °C. The results of the analysis are presented in Figure 6, which shows the performance of the cooling channel.

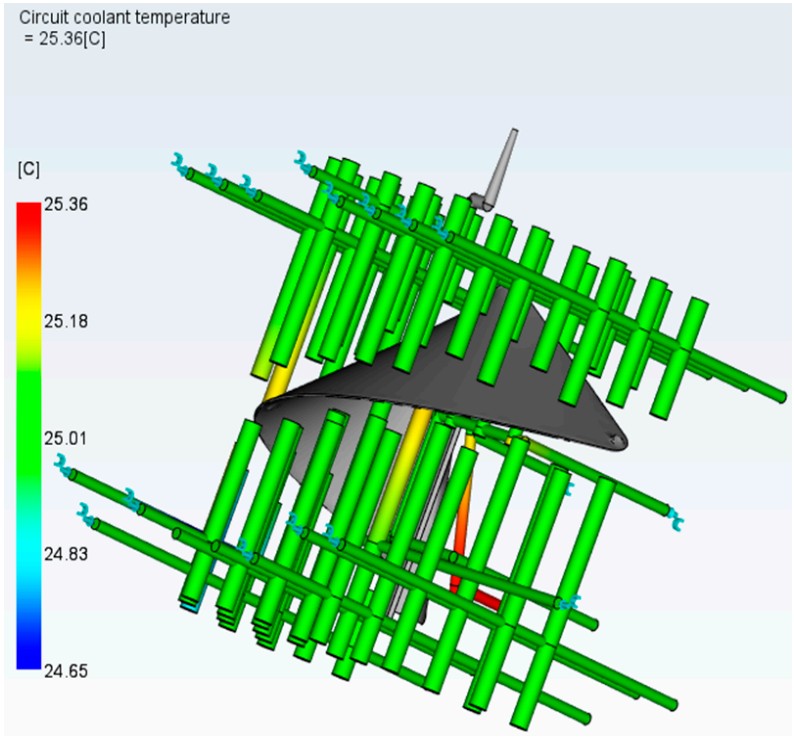

**Figure 6.** Cooling channel performance analysis result.

The lower the temperature of the coolant, the shorter the time it takes to reach the temperature at which the product is ejected, and the cycle time can be reduced. However, if the temperature is too low, temperature deviation can occur in the mold, causing deformation in the product, so it is important to set an appropriate temperature. If the temperature deviation of the coolant is high, deformation of the product is inevitable, and damage to the mold can be caused by the temperature deviation of the mold. Therefore, it is important to check the temperature deviation of the coolant, and generally, the coolant temperature ±2~3 °C is appropriate. The results of the analysis showed that the maximum circuit coolant temperature was 25.36 °C, and the minimum was 24.65 °C. It was concluded that defects related to the cooling performance of both the mold and the parts are within acceptable limits.

To assess whether adequate cooling is achieved during cooling and determine the cooling pump pressure required to maintain cooling efficiency, the circuit Reynolds number analysis and the circuit pressure analysis were conducted. In Moldflow, the default setting for Reynolds number is 10,000, and if it is below 1800, it is considered to be in the laminar region; between 1800 and 2300, it is considered to be in the boundary between laminar and turbulent region, and if it is above 2300, it is considered to be in the turbulent region. Additionally, if the Reynolds number is above 10,000, it is considered that there is sufficient turbulence for adequate cooling. The circuit Reynolds number analysis result showed a maximum of 19,254, and it was judged that cooling is adequately taking place. The circuit pressure analysis predicts the pressure applied to the coolant in the cooling channel during cooling. If the pressure applied to the coolant decreases, the cooling efficiency decreases, and if the pressure applied to the coolant is excessively high, it can cause damage to the mold. Therefore, the analysis was conducted to identify the maintainable pressure and determine the damage to the mold. The result of the circuit pressure analysis showed a maximum of 345.7 kPa, and it was judged that the pump should maintain a pressure of 345.7 kPa, which will not cause damage to the mold.

*4.2. Filling Analysis*

Analyses related to filling were conducted for the evaluation of injection molding characteristics. The filling behavior can be analyzed through Confidence of fill, Fill time, and Quality prediction. The confidence of fill indicates the probability of plastic filling a region within the cavity under conventional injection molding conditions. Figure 7 shows the results of the analysis of the confidence of fill. The analysis confirmed that all three materials were 100% filled. The fill time analysis shows the filling pattern over time using contour lines. Parts on the same contour line are filled at the same time, and the contour lines are shown in different colors. Areas with closer contour lines have a slower filling rate compared to wider areas, and this slow filling rate can cause the temperature of the flow front to drop rapidly, leading to defects or weld lines. Therefore, it is important to understand the filling pattern over time.

Table 5 displays the fill time data estimated, and Figure 8 shows the fill time results for the Supran 1340 materials. The fill time analysis results showed slightly varying fill times, but all three materials showed uniform isothermal lines, as shown in Figure 8. The fill time was the least for RB84HP at 9.441 s and the most for Supran 1340 at 11.82 s. Nevertheless, this difference of approximately 2 s is acceptable.

**Table 5.** Fill time result data.

| Model | Estimated Time (s) |
| --- | --- |
| Supran 1340 | 11.82 |
| RB84HP | 9.441 |
| P7−45FG−0791 | 11.30 |

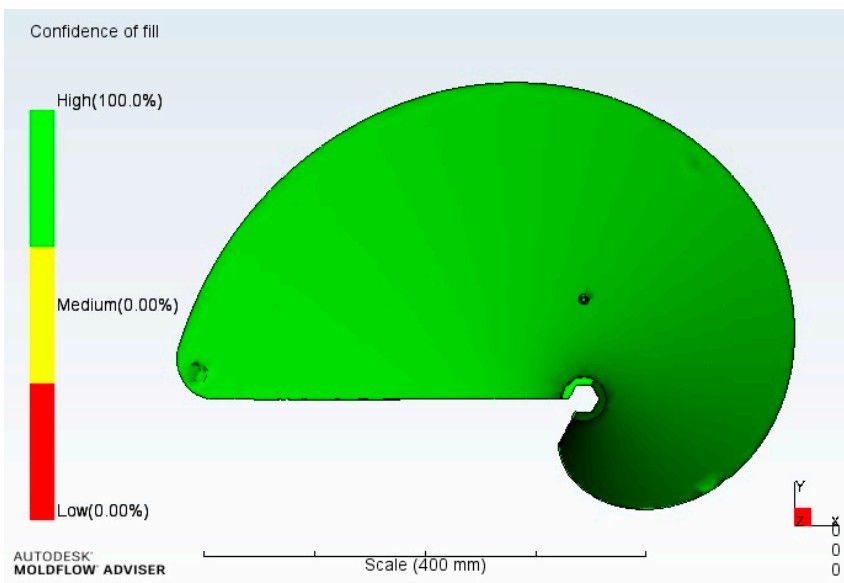

**Figure 7.** Confidence of fill analysis result.

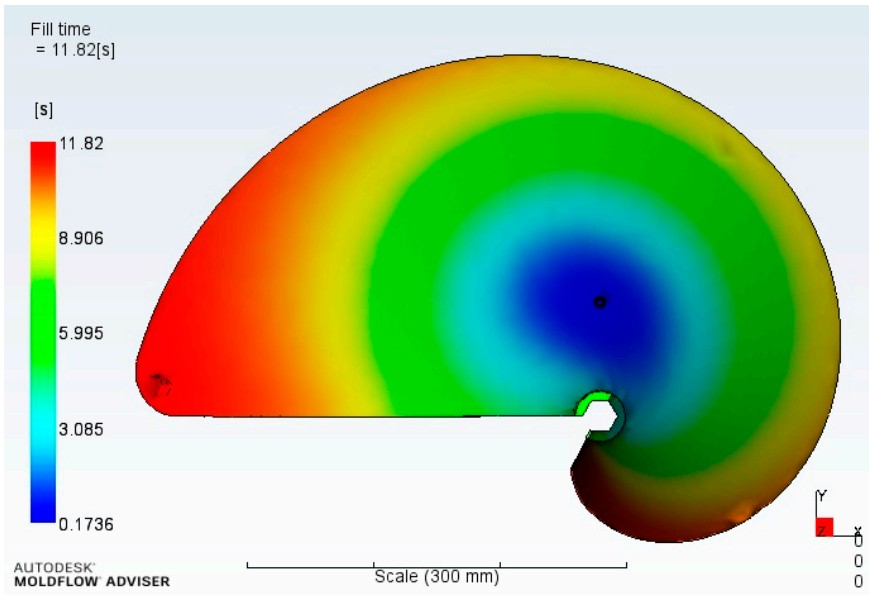

**Figure 8.** Fill time analysis result.

The quality prediction result was used to estimate the quality of mechanical properties and appearance of the part. The contour area was divided into three areas, and it was predicted that the higher the quality area, the better the mechanical properties or appearance quality of the product. Surface defects such as silver streaks, flow marks, and crazing are likely to occur in low-quality parts. Table 6 shows the estimated quality data, and Figure 9 shows the quality prediction results for P7−45FG−0791 materials. The analysis of the quality prediction results shows that 0.22% of the total product is estimated to be low-quality in the P7−45FG−0791 material, while no low-quality parts are expected in the other two materials. The other parts showed similar tendencies, with more than 70% of each of the three materials predicted to be high-quality, indicating that the overall product quality is likely to be good.

**Table 6.** Estimated time result data.

| Model | Estimated Quality (%) | | |
|:---:|:---:|:---:|:---:|
| | High | Medium | Low |
| Supran 1340 | 74.7 | 25.3 | 0.00 |
| RB84HP | 75.3 | 24.7 | 0.00 |
| P7−45FG−0791 | 74.6 | 25.2 | 0.22 |

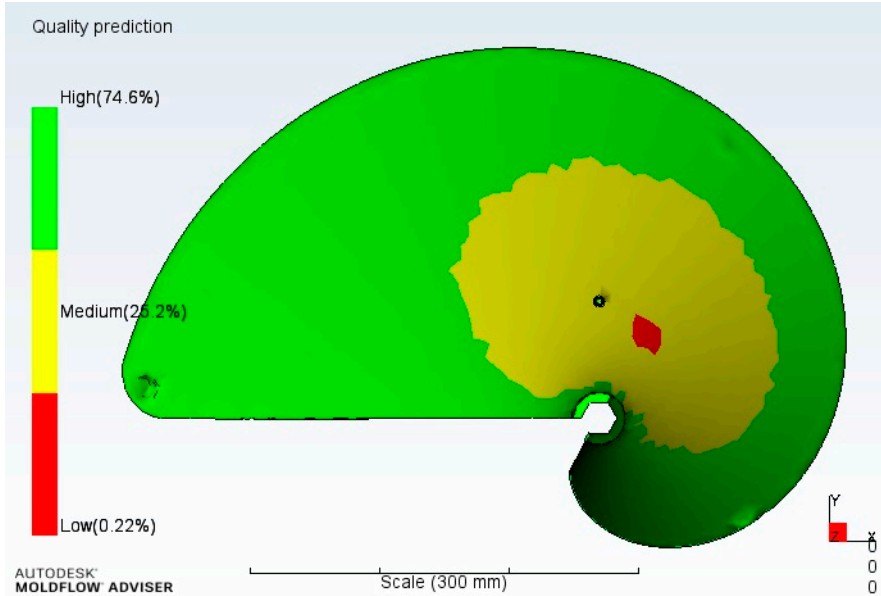

**Figure 9.** Quality prediction analysis result.

*4.3. Cooling Analysis*

In order to determine the temperature of the melt front in the flow lines during injection molding, as well as the residual heat state of the product and the time required to reach ejection temperature, various analyses were conducted. Specifically, this session included an analysis of the time required to reach ejection temperature, the average temperature of the part, and the cooling quality. The time required to reach ejection temperature refers to the period from the beginning of filling until the product has cooled down to 80–90% of the total cooling time. Table 7 displays the duration needed to reach the ejection temperature, while Figure 10 exhibits the corresponding results for Supran 1340 materials, presented with a temperature distribution.

**Table 7.** Estimated time for reaching the ejection temperature.

| Feedstock | Estimated Time (s) |
|:---:|:---:|
| Supran 1340 | 235.5 |
| RB84HP | 115 |
| P7−45FG−0791 | 211.1 |

Based on the analysis results of the time required to reach the ejection temperature, each material showed a different time to reach the ejection temperature. However, as shown in Figure 10, they exhibited similar tendencies. The time to reach the temperature at which ejection is possible was the shortest for RB84HP at 115 s, while Supran 1340 and P7−45FG−0791 showed times of 235.5 s and 211.1 s, respectively. It was concluded that the presence or absence of glass fiber affected the results.

The Average temperature analysis represents the average temperature in the thickness direction at the point where the product is 100% filled. A larger difference between the maximum and minimum average temperatures leads to more thermal deformation in the product shape due to temperature deviation after ejection. Table 8 presents the average temperature data.

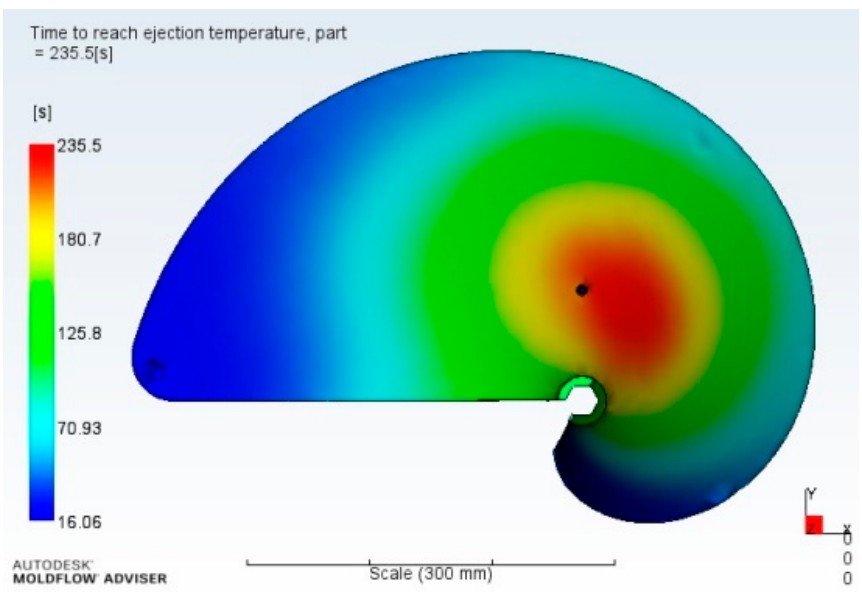

**Figure 10.** Time to reach ejection temperature analysis result.

**Table 8.** Average temperature at complete fill.

| Model | Max. Temp. (°C) | Min. Temp. (°C) | Temp. Difference (°C) |
|---|---|---|---|
| Supran 1340 | 247.7 | 188.1 | 59.6 |
| RB84HP | 215.6 | 142.6 | 73 |
| P7−45FG−0791 | 235.3 | 162.1 | 73.2 |

The results of the average temperature analysis showed temperature differences, but all three materials exhibited a similar trend. Supran 1340 showed the smallest temperature difference at 59.6 °C, while RB84HP and P7−45FG−0791 exhibited temperature differences of 73 °C and 73.2 °C, respectively.

The analysis of cooling quality represents the tendency of residual heat remaining in the product due to its shape and thickness. It is predicted that a large amount of heat residue may result in low cooling quality, increasing the likelihood of severe deformation of the product. Table 9 shows the cooling quality data, and Figure 11 illustrates the cooling quality analysis results for P7−45FG−0791 materials.

**Table 9.** Cooling quality result data.

| Model | Cooling Quality (%) | | |
|---|---|---|---|
| | High | Medium | Low |
| Supran 1340 | 71.0 | 25.3 | 3.67 |
| RB84HP | 74.4 | 22.3 | 3.24 |
| P7−45FG−0791 | 69.7 | 26.2 | 4.07 |

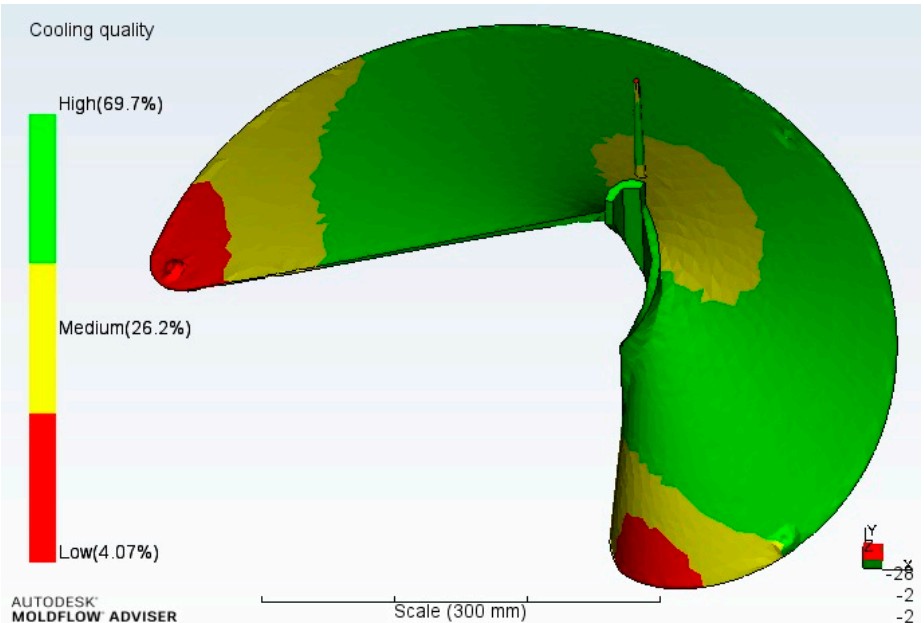

**Figure 11.** Cooling quality analysis result.

The cooling quality analysis revealed a slight difference in quality ratios among the three materials, but they all showed a similar trend to Figure 11. Low-quality regions were predicted to appear at the blade tip and spiral end of the structure, and deformation was expected in areas with a high amount of residual heat. However, all three materials exhibited high-quality areas of around 70% or more, with the middle and high-quality sections accounting for over 90%, indicating minimal deformation.

### 4.4. Injection Pressure Analysis

The injection pressure analysis represents the pressure applied at the nozzle end of the sprue just before the pressure/velocity change occurs during the packing stage of the product molding. This is typically when the maximum pressure is applied during the injection process, and it provides a reference for determining the necessary pressure for injection molding. The pressure value can be used as a reference for selecting mold materials and injection molding machines or as a basis for modifying the model if excessive pressure is expected. Table 10 presents the data for the maximum injection pressure.

**Table 10.** Time data for reaching the extraction temperature.

| Model | Max. Injection Pressure (MPa) |
| :---: | :---: |
| Supran 1340 | 13.91 |
| R84HP | 33.04 |
| P7−45FG−0791 | 38.52 |

The injection pressure analysis showed some differences in pressure, but all three materials exhibited a similar trend. Supran 1340 had the lowest injection pressure at 13.91 MPa, while RB84HP and P7−45FG−0791 had 33.04 MPa and 38.52 MPa, respectively. The lower the predicted maximum injection pressure, the less burden on the injection molding machine and mold, indicating that Supran 1340 is expected to provide the smoothest injection molding.

### 4.5. Defects Analysis

To predict potential defects in products after injection molding, Moldflow injection molding analyses were conducted in this study. Specifically, the analyses included fiber

orientation, orientation at skin, shrinkage, estimated sink mark, deflection and warpage, and air traps and weld lines.

The fiber orientation and the orientation at skin analysis provides valuable insight into the molecular orientation on the outer surface of the part, indicating the average principal alignment direction for the local area at the end of filling. This is particularly important for materials containing glass fibers, as shrinkage rate and mechanical strength vary depending on orientation. Figure 12 displays the fiber orientation and the orientation at skin analysis results for the materials. Figure 12a,b are the fiber orientation analysis results by direction of Supan 1340 material and P7−45FG−0791 material, and Figure 12c is the orientation at skin analysis result of RB84HP material. The results of the orientation at the skin analysis showed similar trends for all three materials, as seen in Figure 12. The analysis predicted that the glass fibers would be oriented in the direction perpendicular to the direction in which the blades would be bent by wind pressure. The orientation was found to be good, as there was no random orientation due to turbulence. It was determined that this would increase the mechanical strength of the blade.

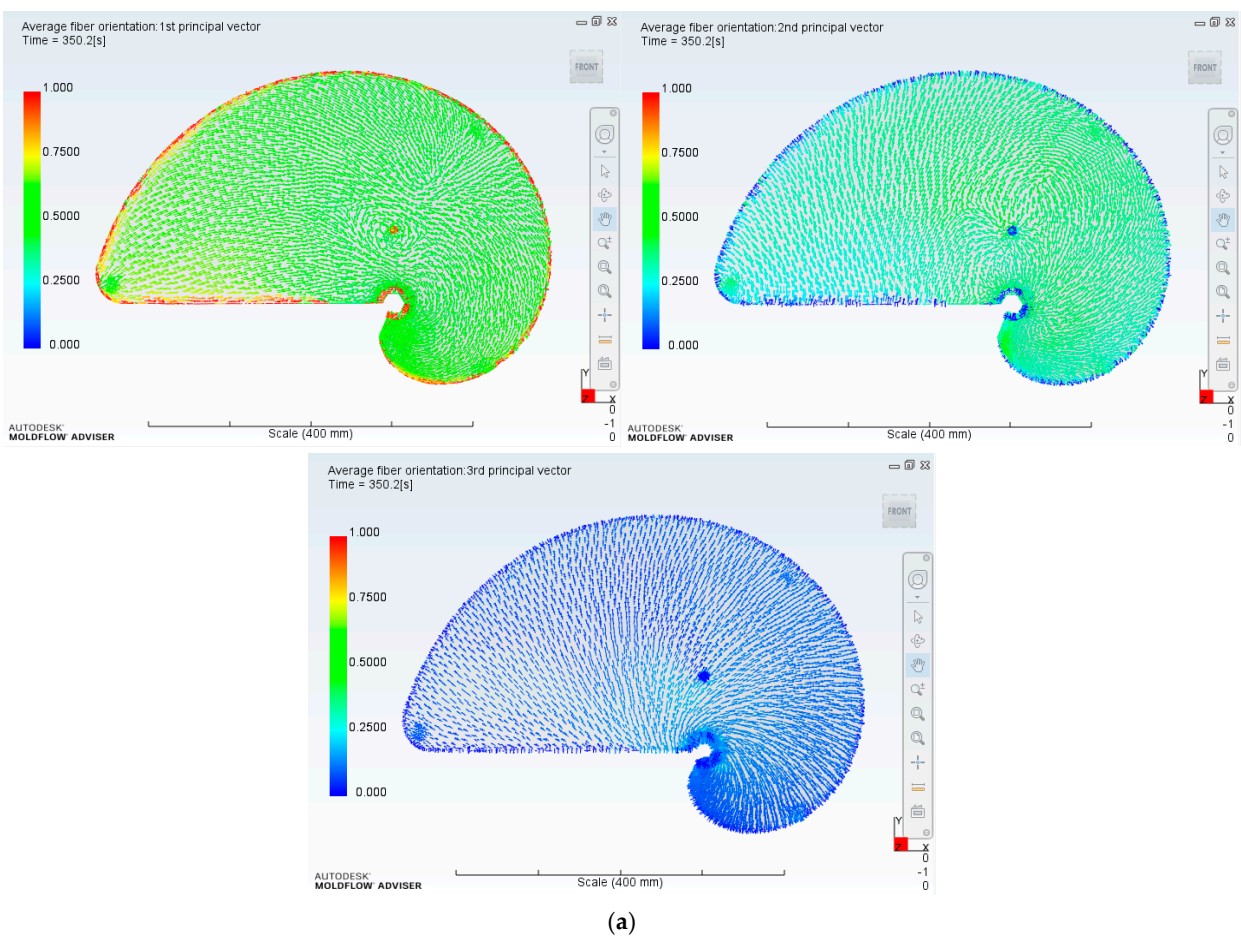

(**a**)

**Figure 12.** *Cont*.

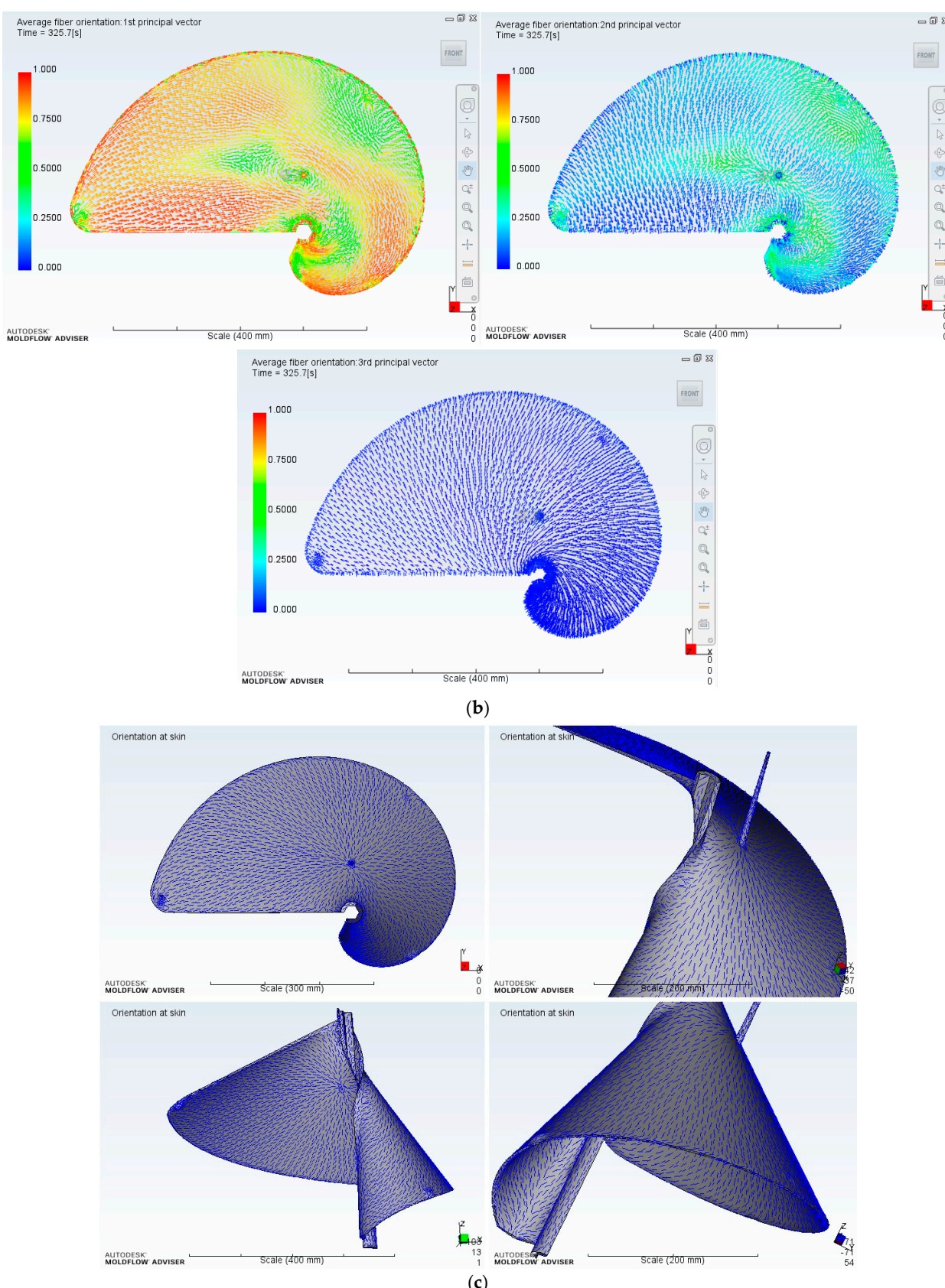

**Figure 12.** Fiber orientation results by direction and orientation at skin analysis results. (**a**) Fiber orientation results by direction of Supan 1340 material. (**b**) Fiber orientation results by direction of P7−45FG−0791 material. (**c**) Orientation at skin results of RB84HP material.

After injection molding, Moldflow analysis was performed to predict the volume shrinkage rate of the product. The volumetric shrinkage analysis shows the estimated volumetric shrinkage of the product when the resin is cooled to room temperature. The volumetric shrinkage is determined by the temperature and pressure of each part of the packing pressure and cooling process. In general, the faster the cooling rate, the higher the packing pressure, and the longer the packing pressure time, the lower the volume shrinkage. The volumetric shrinkage affects the dimensions, deformation, and sink mark of the product, and if the differential shrinkage is large, the possibility of contraction deformation increases. Table 11 presents the volumetric shrinkage results, and Figure 13 shows the volumetric shrinkage analysis results for Supran 1340 materials. The maximum volumetric shrinkage showed 15.53% of Supran 1340 materials, 14.86% of RB84HP materials, and 14.14% of P7−45FG−0791 materials. All three materials have the highest volumetric shrinkage around the gate, as shown in Figure 13, and the volumetric shrinkage decreased towards the end of the product.

**Table 11.** Volumetric shrinkage analysis results.

| Model | Estimated Max. Volumetric Shrinkage (%) |
|---|---|
| Supran 1340 | 15.53 |
| RB84HP | 14.86 |
| P7−45FG−0791 | 14.14 |

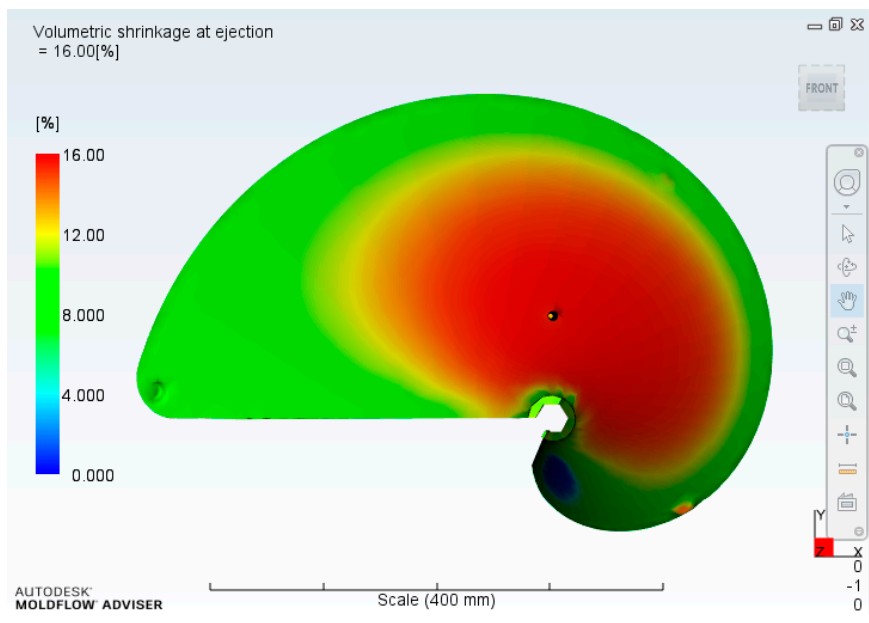

**Figure 13.** Volumetric shrinkage analysis results.

To predict a common injection molding defect, sink mark, Moldflow analysis was conducted. A sink mark is a sunken area on the surface of a product near ribs or bosses caused by inadequate packing pressure far from the gate. It is a critical factor to be controlled since it affects the thickness of the product in the depth direction. If sink marks are observed to be significant, one possible remedy is to add additional gate locations or increase the thickness of the product while reducing the thickness of ribs or bosses, which can help to improve packing near the critical areas and reduce the occurrence of sink marks. Table 12 presents the sink mark results, and Figure 14 shows the sink mark analysis results for Supran 1340 materials. The sink mark analysis results showed some numerical differences among the three materials, but as shown in Figure 14, they all appeared at

the joint between the blade and shaft assembly. The size of the sink marks for all three materials was judged to be 0.5 mm or less, which would not affect the product.

**Table 12.** Estimated sink mark analysis results.

| Model | Estimated Max. Sink Marks (mm) |
|---|---|
| Supran 1340 | 0.4680 |
| RB84HP | 0.3976 |
| P7−45FG−0791 | 0.3966 |

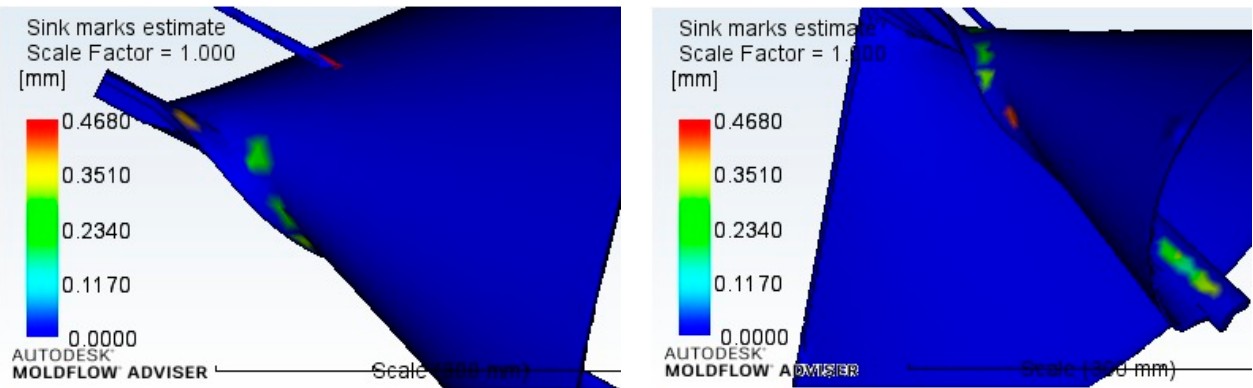

**Figure 14.** Location and size of the sink mark analysis results.

The deflection and warpage analysis results demonstrate how the part may deform from the original design due to various factors, including the shrinkage factor of the injection material, the shape of the product, and differences in residual and cooling temperatures. The analysis results for Supran 1340, P7−45FG−0791, and RB84HP are presented in Tables 13 and 14 and Figure 15. While RB84HP exhibited the highest deflection of 9.273, it had the lowest warpage of 2.96%. However, since the deflection and warpage were predicted to occur at the connection point of the shaft and blade, it is expected to be critical during the product assembly. Conversely, for Supran 1340 and P7−45FG−0791, it was judged that there would be little deformation in the shaft and fixing rod, but there would be deformation at the end of the blade.

**Table 13.** Deflection results data.

| Model | Estimated Max. Deflection (mm) |
|---|---|
| Supran 1340 | 8.340 |
| RB84HP | 9.273 |
| P7−45FG−0791 | 8.088 |

**Table 14.** Warpage result data.

| Model | Warpage Indicator (%) | | |
|---|---|---|---|
| | High | Medium | Low |
| Supran 1340 | 4.34 | 9.39 | 86.3 |
| RB84HP | 2.96 | 14.7 | 82.3 |
| P7−45FG−0791 | 5.51 | 6.74 | 87.8 |

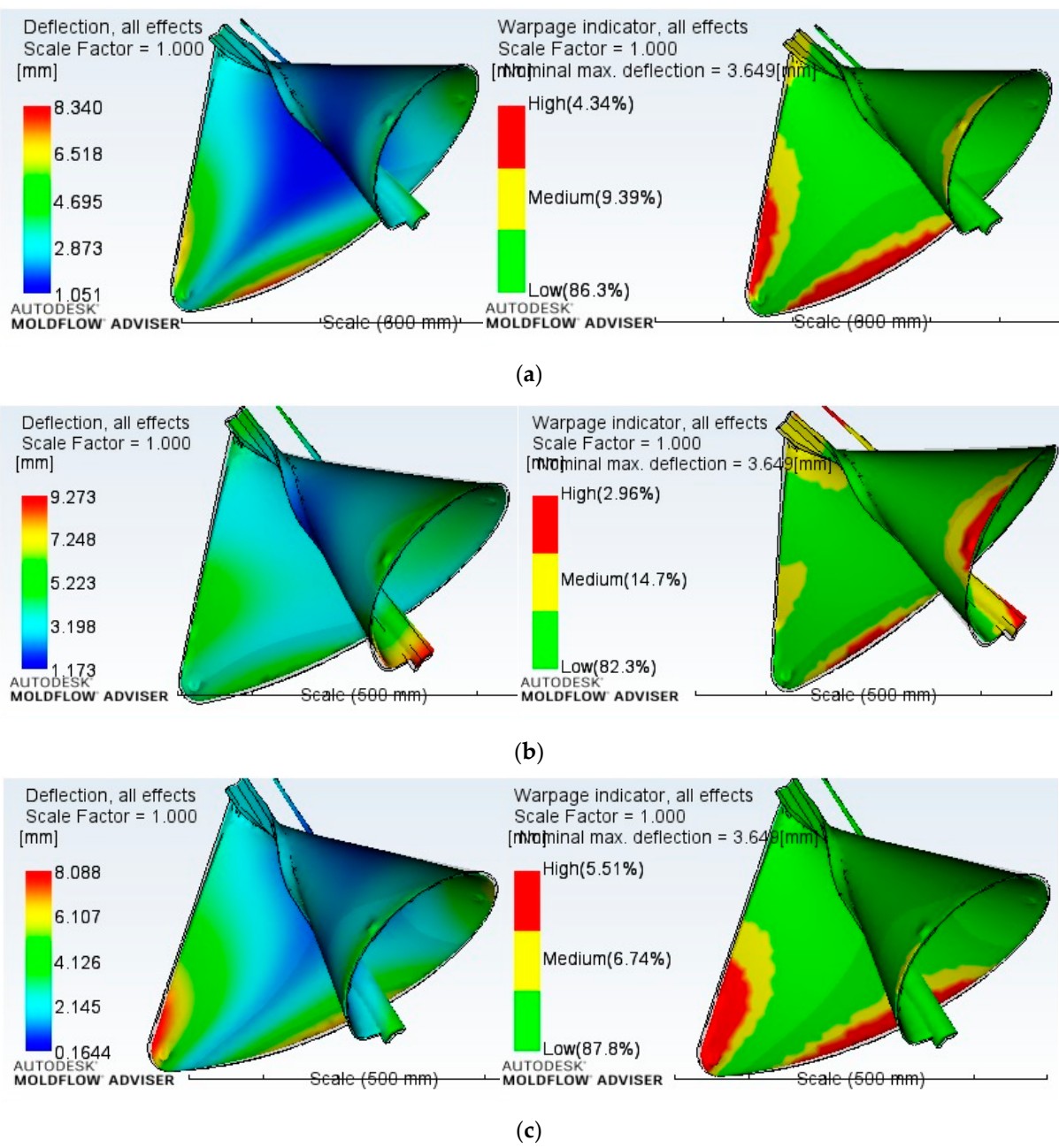

**Figure 15.** Deflection and warpage analysis results. (**a**) Supran 1340; (**b**) RB84HP; (**c**) P7−45FG−0791.

Air traps are a defect that occurs when air or volatile gases from the resin cannot be released from the cavity during injection molding, resulting in an internal void. This can be eliminated by identifying the location and installing appropriate air vents. Weld lines are caused by the meeting of multiple flow fronts due to the shape of the product and the number of gates, resulting in weakened mechanical strength and an uneven surface. To minimize weld lines, gates should be designed to avoid overlap with stress points, and their number and length should be minimized. The analysis of air traps and weld lines for Supran 1340 materials showed similar results for all three materials, with air traps occurring at the end of the blade and shaft and weld lines occurring near the injection gate. The high temperatures around the injection gate were found to minimize the formation of weld lines and thus have no effect on the actual part. Figure 16 illustrates these findings.

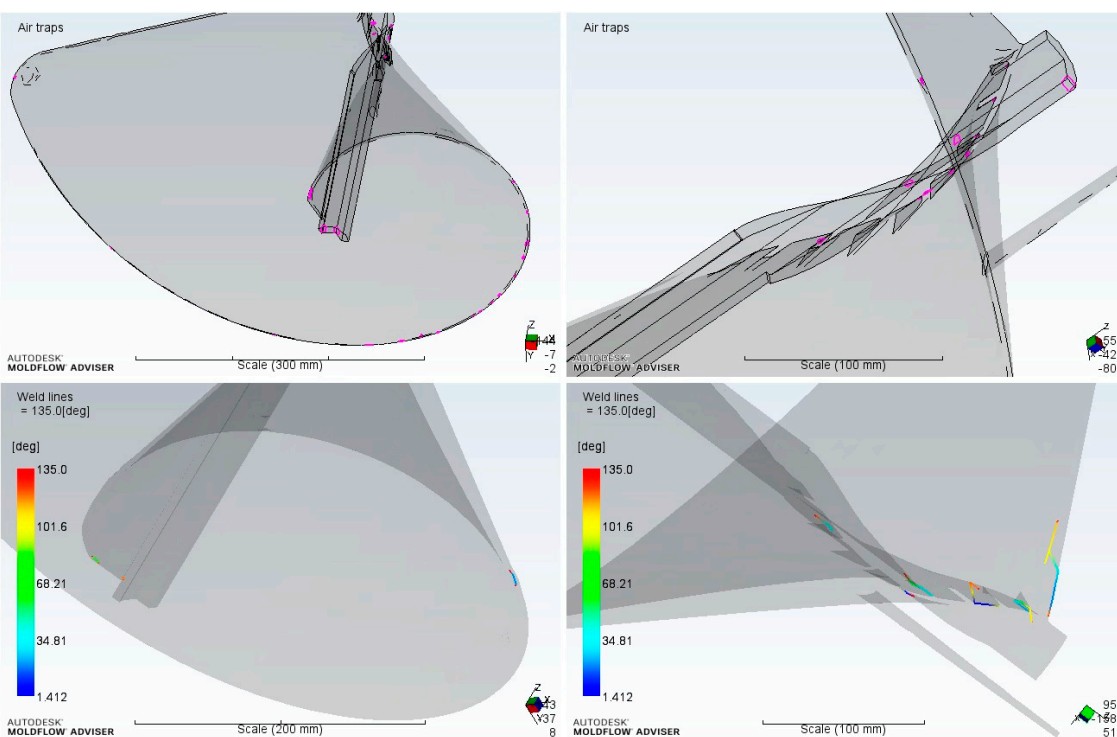

**Figure 16.** Result of the analysis of air traps and weld lines.

### 4.6. Tensile and Bending Tests for Material Selection

Tensile and bending tests were conducted using a universal testing machine [25] to finalize the material selection. RB84HP, which was deemed to have critical assembly defects and relatively low mechanical properties, was excluded, and two materials, Supran 1340 and TEKAFINTM P730G45, were tested. As TEKAFINTM P730G45 properties were not provided in the Moldflow library, P7−45FG−0791 was used as a substitute for analysis. However, to obtain more accurate data and select the material, actual materials were injected to create tensile and bending specimens for testing. Five specimens were tested for each material, and the test results were averaged.

Tensile and bending tests were carried out using RB 301 UNITECH-M universal testing machine (R&B Co., Daejeon, Republic of Korea). Tensile tests were performed using specimens made according to ASTM D638 specifications with a displacement control of 50 mm/min while bending tests were conducted using specimens made according to ASTM D790 with a displacement control of 10 mm/min. The results are presented in Table 15. The tensile test showed that Supran 1340 had a yield strength that was 6.2% higher, a tensile strength that was 1.46% higher, and a 19.8% higher elongation at break than TEKAFINNTM P730G45. Additionally, the bending test showed that Supran 1340 had a 10% higher flexural strength and a 44% higher flexural modulus than TEKAFINNTM P730G45. Therefore, Supran 1340 was selected as the final injection molding material for its excellent mechanical properties.

**Table 15.** Tensile and bending test results data.

| Items | Material Types | |
|---|---|---|
| | **Supran 1340** | **TEKAFINTM P730G45** |
| Tensile strength (MPa) | 90.2 | 88.9 |
| Yield strength (MPa) | 77.9 | 73.0 |
| Elongation (%) | 1.15 | 0.96 |
| Flexural strength (MPa) | 114.9 | 103.4 |
| Flexural modulus (GPa) | 7.5 | 5.2 |

## 5. Test Injection Molding and Assembly

### 5.1. Mold Making

A mold was created specifically for testing injection molding. The design of the mold was based on the specified mold data, and it was constructed using S55C steel. Figure 17 displays a picture of the completed mold.

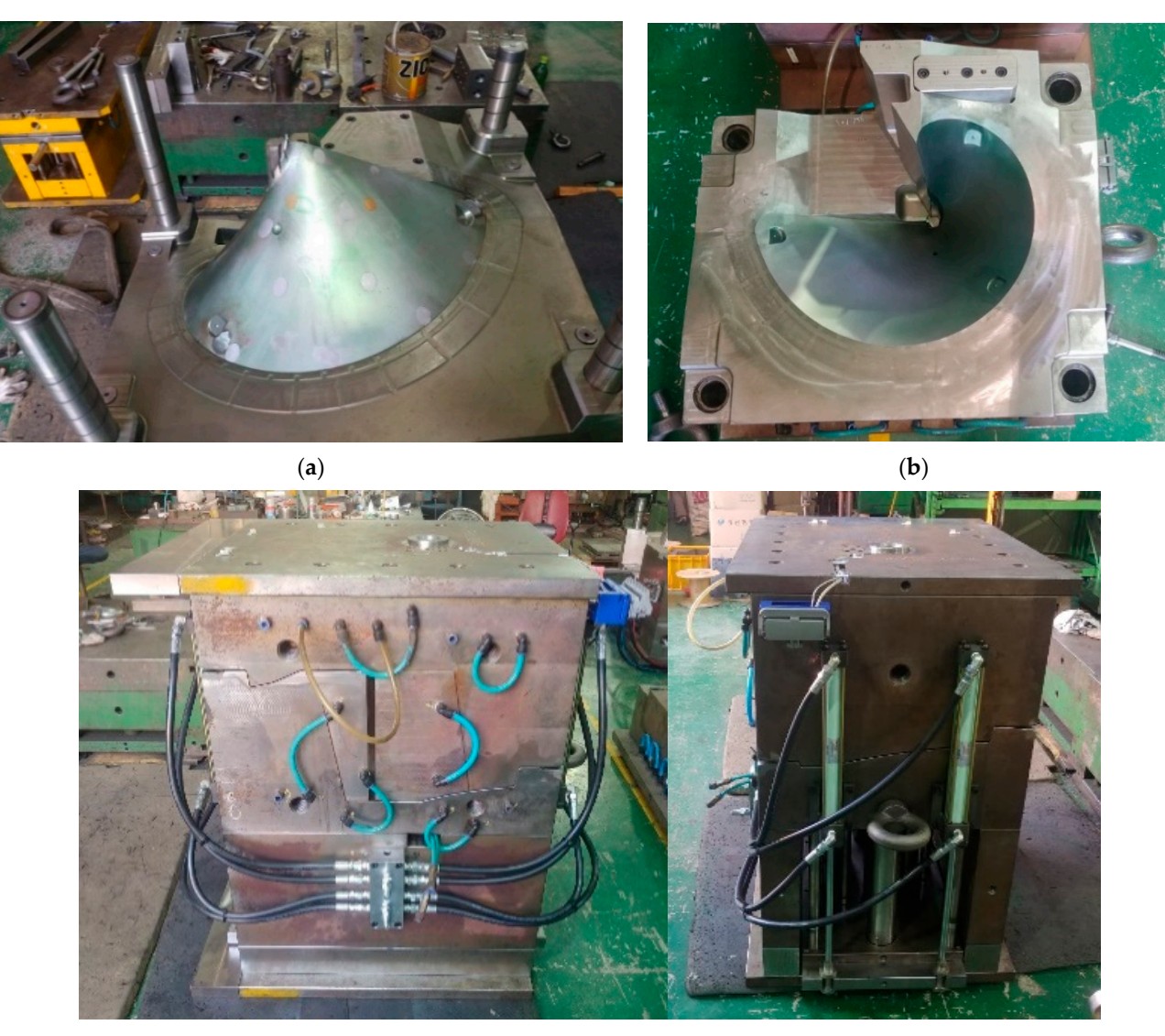

(a)

(b)

(c)

**Figure 17.** Photographs of the manufactured mold. (**a**) Mold movable side; (**b**) Mold fixed side; (**c**) Mold assembly with cooling channels.

## 5.2. Test Injection Molding

The manufactured mold was securely attached to an injection molding machine, and a test injection molding was carried out using the UBEMAX-ST series 1300-280 injection molding machine (UBE Co., Okinoyama, Japan). The injection molding conditions are presented in Tables 16–20. The cylinder temperature was set to 230 °C for NH-H3, 220 °C for H4, and 210 °C for H5. For metering conditions, a pressure of 100 MPa, a velocity of 50%, a distance of 190 mm, and a suck back of 8 mm was applied. Water was used as a refrigerant for cooling, and the temperature was set to 30 °C. The average pressure and velocity during injection were 8 MPa and 40%, respectively. Packing pressure 1 was 5 MPa for 1 s, packing pressure 2 was 3 MPa for 15 s, and packing pressure 3 was 2 MPa for 35 s, with a total of 51 s. The back pressure was 1 MPa, and the cooling time was set to 180 s.

**Table 16.** Input data for cylinder temperature settings.

| Items | Numerical Values | | | | | |
|---|---|---|---|---|---|---|
| Heater number | NH | H1 | H2 | H3 | H4 | H5 |
| Cylinder temperature (°C) | 230 | 230 | 230 | 230 | 220 | 210 |

**Table 17.** Input data for Feeding conditions.

| Items | Feeding Conditions |
|---|---|
| Pressure (MPa) | 100 |
| Velocity (%) | 50 |
| Distance (mm) | 190 |
| Suck back (mm) | 8 |

**Table 18.** Input data for injection conditions.

| Items | Injection Conditions |
|---|---|
| Average pressure (MPa) | 8 |
| Velocity (%) | 40 |

**Table 19.** Input data for packing conditions.

| Items | Pressure (MPa) | Time (s) |
|---|---|---|
| Packing pressure 1 | 5 | 1 |
| Packing pressure 2 | 3 | 15 |
| Packing pressure 3 | 2 | 35 |

**Table 20.** Input data for back pressure and cooling time conditions.

| Items | Conditions |
|---|---|
| Back pressure (MPa) | 1 |
| Cooling time (s) | 180 |

As a result of the test injection molding, the cycle time was 195 s, and the weight of the product was 2020 g. Figure 18 shows the blade manufactured by the test injection molding.

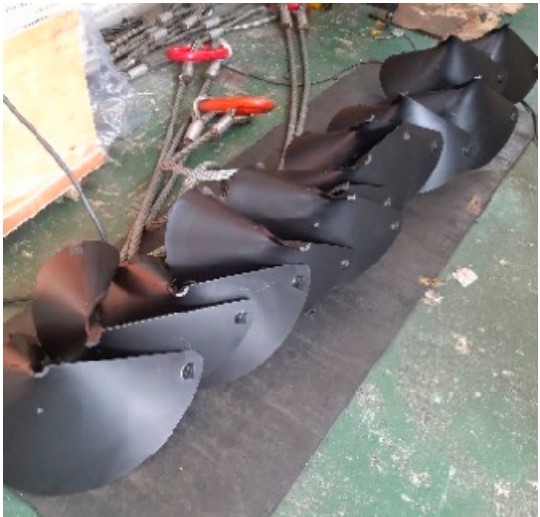
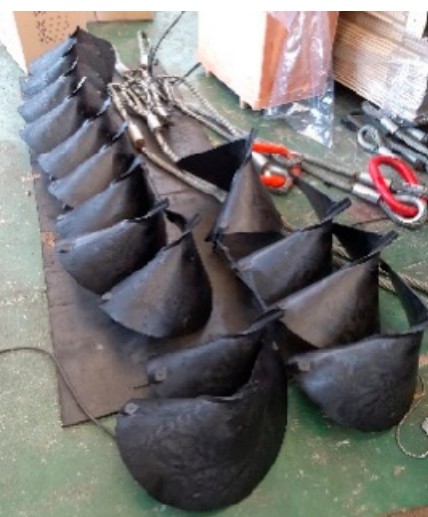

**Figure 18.** Blades manufactured by test injection molding.

*5.3. Small Wind Turbine Assembly*

The spiral small wind turbine was constructed using a blade that was manufactured through injection molding. Figure 19 illustrates the assembly of the spiral small wind turbine, and it was verified that the injection molded product was assembled seamlessly. Following the assembly, the turbine was installed, as shown in Figure 20, and its operational performance, blade damage, and stability were observed for a certain period of time. It was confirmed that the wind power generation was safely carried out without any damage.

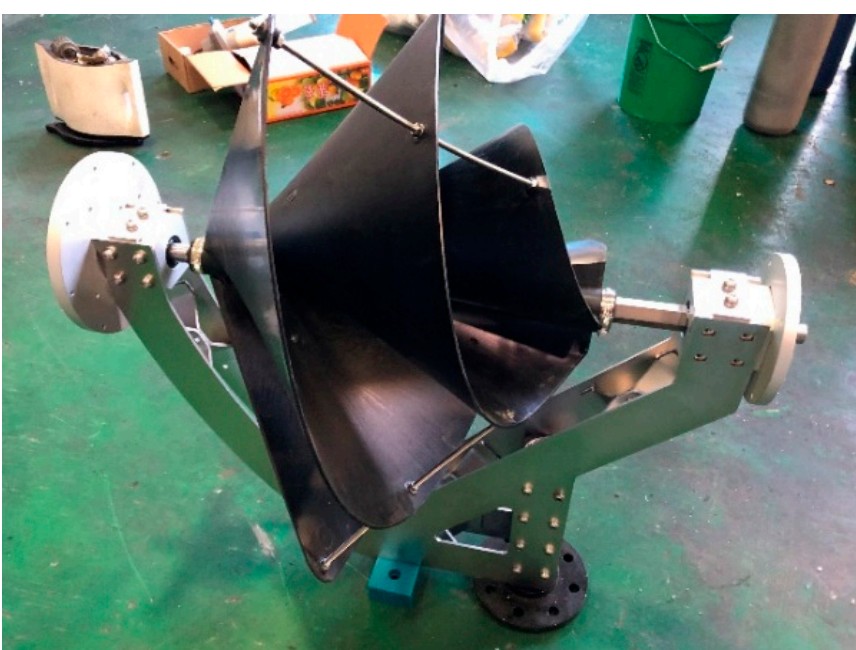

**Figure 19.** Spiral small wind turbine assembly with injection molded blades.

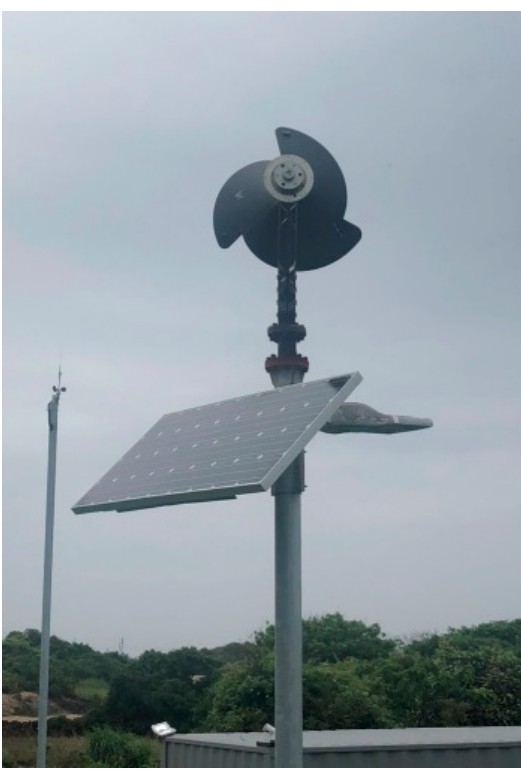

**Figure 20.** Spiral small wind turbine under operation.

### 6. Conclusions

This paper presents a study on the mass production of the blade of a spiral small wind turbine using injection molding. The blade was divided based on its shape, and the mold was designed accordingly. Moldflow injection molding analysis was conducted to evaluate the feasibility of injection molding and to select suitable injection materials.

Prior to injection molding, the product's shape was modeled, and a cooling system was designed. Preprocessing was performed for injection molding analysis, and data for three selected materials were input. Injection molding analysis was conducted for each material. The mechanical properties of the materials were derived through tensile and bending tests, and Supran 1340 was selected as the final injection material based on the analysis of the data and mechanical properties.

To determine the productivity of actual injection molding and identify any product abnormalities, Moldflow analysis for defects was performed, and test injection molding was conducted using a UBEMAX-ST series 1300-280 injection machine. The selected Supran 1340 was used to manufacture a small wind turbine blade with a cycle time of 195 s and a weight of 2020 g. The assembly of the test injection molded blade with other parts was checked, and it was confirmed that there were no defects in the assembly and the blades were functional.

Finally, an assembled small wind turbine was installed to observe the operation of the blade. The product was visually determined for damage, and the operation was checked by observing the generation of electricity. Therefore, we confirmed that power is produced stably without damage.

Our study is a critical step in the mass production of the product once it is developed. The division of the complex geometry of the spiral blades and the analysis of the material selection method and data will be of great interest to injection molding engineers. Furthermore, our work will aid the small wind turbine sector by providing guidelines for mass-producing blades with complex shapes to scholars and engineers studying wind turbines.

It is worth noting that our study specifically focused on the possibility of utilizing injection molding as a means of mass-producing small wind turbine blades. This study is

particularly significant because it demonstrates that the injection molding method can be utilized to manufacture blades with complex geometries, which diverges from traditional blade manufacturing techniques. We believe that our findings will contribute to the development of more efficient and cost-effective manufacturing processes for small wind turbine blades.

**Author Contributions:** Conceptualization, J.K., H.J. and S.A.; methodology, J.K., D.J. and U.T.; software, J.K. and U.T.; validation, J.K., D.J., H.J. and S.A.; formal analysis, J.K., D.J. and U.T.; investigation, J.K., U.T., M.M. and J.B.; resources, J.K., D.J., M.M. and J.B.; data curation, J.K., H.J. and J.B.; writing—original draft preparation, J.K., H.J. and S.A.; writing—review and editing, J.K., U.T., H.J. and S.A.; supervision, H.J. and S.A.; project administration, J.K., J.B., H.J. and S.A.; funding acquisition, M.M., J.B., H.J. and S.A. All authors have read and agreed to the published version of the manuscript.

**Funding:** This work was supported by the Energy Core Technology Program of the Korea Institute of Energy Technology Evaluation and Planning (KETEP), which was granted financial resources from the Ministry of Trade, Industry & Energy, Republic of Korea (No. 20183030029120). This research was supported by Korea Basic Science Institute (National Research Facilities and Equipment Center) grant funded by the Ministry of Education (grant No. 2021R1A6C101A449).

**Institutional Review Board Statement:** Not applicable.

**Informed Consent Statement:** Not applicable.

**Data Availability Statement:** The data presented in this study are openly available in section 4.6 at [doi: 10.14775/ksmpe.2020.19.02.038], reference number [25].

**Conflicts of Interest:** The authors declare no conflict of interest.

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
