# Peer review of "A Feasibility Study on the Use of Injection Molding Systems for Mass Production of 100W Class Wind Turbine Blades"

_processes, doi:10.3390/pr11061855_

Round 1
Reviewer 1 Report
The article is very well presented. The version of Moldflow software (Adviser or Insight) must be mentioned. Consistency in writing Moldflow should be maintained throughout the article [In some places it is mentioned as Mold-flow]
Author Response
Response to Reviewer 1 Comments
The article is very well presented. The version of Moldflow software (Adviser or Insight) must be mentioned. Consistency in writing Moldflow should be maintained throughout the article [In some places it is mentioned as Mold-flow]
Thank you for your good evaluation. We edited the article as below after accepting your advice.
Point 1: The version of Moldflow software (Adviser or Insight) must be mentioned.
Response 1: We changed ‘Moldflow’ in line 121 to ‘Moldflow Advisor 2018’.
Point 2: Consistency in writing Moldflow should be maintained throughout the article [In some places it is mentioned as Mold-flow]
Response 2: We changed ‘mold-flow’ to ‘moldflow’.

Reviewer 2 Report
1. Figure 1 should be of better quality.
2. Literature introduction lacking in literature. It is worth analyzing in terms of Predicting effect of fiber orientation on chosen strength properties, Determination of viscosity curve and pVT properties.
3. All units should be described in the SI system.
4. Error - figure 2 (line 141) is figure 4 - the quality of this screenshot should be improved.
5. Which fiber orientation prediction model was chosen - Tucker-Folgar or RSC?
6. On what basis was the injection point selected?
7. Why in table 4. Injection time selected and Velocity/pressure switch-over selected as automatic?
8. Why Dual Domain Mesh and not 3D?
9. Figure 7-15 - Poor Screen Quality.
10. No chapter discussing results and comparing it with other publications.
11. Were the actual injection tests statistically processed?
12. The work contains a lot of editing errors - it requires thorough improvement.
13. In my opinion, it is important to show the results of the fiber orientation simulation in the work - the resultant and the 1st, 2nd and 3rd order.
14. What version of Modflow was used?
15. What type of analysis was chosen? E.g. Fill+Pack+Cool+Warp?
16. It is worth including the pVT curve from the Tait model in the work (it is in the material base of the polymer composite used).
17. Figure 12 - is it really a fiber orientation simulation because I'm not sure?.. please check...
18. What was the shrinkage of the simulation product?
Author Response
Response to Reviewer 2 Comments
Thank you for your professional evaluation. We accepted the advice of Reviewer and revised the article as below, and wrote the answer as below.
Point 1: Figure 1 should be of better quality.
Response 1: The quality of Figure 1 has been increased and enlarged.
Point 2: Literature introduction lacking in literature. It is worth analyzing in terms of Predicting effect of fiber orientation on chosen strength properties, Determination of viscosity curve and pVT properties.
Response 2: Figure 5 shows the addition of the 2-states Tiet model pVT curve, and the contents of lines 171-173 are modified.
Before the change
‘The viscosity-shear rate graph of Supran 1340 material used in the analysis is dis-played in Fig. 5, demonstrating viscous thinning behavior. Similarly, RB84HP and P7-45FG-0791 also exhibit similar trends.’
After the change
‘Figure 5 shows The visibility-shear rate graph and 2-states Tiet model pVT curve of Supran 1340 material. Supran 1340 material shows demonstrating viscous thinning be-havior and is identified as a crystalline resin. Similarly, RB84HP and P7-45FG-0791 also exhibit similar trends.’
Point 3: All units should be described in the SI system.
Response 3: The paper was written in compliance with the SI unit.
Point 4: Error - figure 2 (line 141) is figure 4 - the quality of this screenshot should be improved.
Response 4: Figure 2 was changed to Figure 4, and the quality of the picture was improved and enlarged.
Point 5: Which fiber orientation prediction model was chosen - Tucker-Folgar or RSC?
Response 5: The Folgar-Tucker model was chosen.
Point 6: On what basis was the injection point selected?
Response 6: The gate location was selected by the mold engineer's empirical know-how. Added to line 131 as 'The gate location was selected by the mold engineer's empirical know-how.'.
Point 7: Why in table 4. Injection time selected and Velocity/pressure switch-over selected as automatic?
Response 7: This research was carried out to study the appropriate injection time to achieve quality injection molding. The test data were not available for the both the injection time and V/P switchover point. Therefore a default setting were selected. Generally the default V/P switchover point is set to 99% of the mold cavity in moldFlow advisor.
Point 8: Why Dual Domain Mesh and not 3D?
Response 8: Dual Domain mesh uses surface information while 3D Mesh uses information throughout the thickness of the part. Usually Dual Domain Meshes are good enough to use for the simulation where the parts are thin like sheet. In this work, the spiral blades used for injection modling simulation has low thickness compared to the span of the entire blade. Therefore, we used Dual Domain Mesh for this study. However, the use of 3D Mesh can give more accurate warpage results and we would consider the reviewer’s valuable suggestion for our future work.
Point 9: Figure 7-15 - Poor Screen Quality.
Response 9: Figures 7-15 have been changed to improve quality.
Point 10: No chapter discussing results and comparing it with other publications.
Response 10: We added the following information to the 539-551 line.
‘Our study is a critical step in the mass production of the product once it is developed. The division of the complex geometry of the spiral blades and the analysis of the material selection method and data will be of great interest to injection molding engineers. Fur-thermore, our work will aid the small wind turbine sector by providing guidelines for mass-producing blades with complex shapes to scholars and engineers studying wind turbines.
It is worth noting that our study specifically focused on the possibility of utilizing injection molding as a means of mass-producing small wind turbine blades. This study is particularly significant because it demonstrates that the injection molding method can be utilized to manufacture blades with complex geometries, which diverges from traditional blade manufacturing techniques. We believe that our findings will contribute to the de-velopment of more efficient and cost-effective manufacturing processes for small wind turbine blades.’
Point 11: Were the actual injection tests statistically processed?
Response 11: The actual injection tests were carried out in the following process.
- Clamp the mold to the injector.
- Check the ejecting system at 5 times idle.
- Heat the mold to the proper temperature.
- Conduct 5 injection cycles to test injection conditions.
- After that, we evaluated the product and started the actual injection tests.
Point 12: The work contains a lot of editing errors - it requires thorough improvement.
Response 12: We took your advice and edited the error as much as possible.
Point 13: In my opinion, it is important to show the results of the fiber orientation simulation in the work - the resultant and the 1st, 2nd and 3rd order.
Response 13: As advised by Reviewer,we added the results of The fiber orientation of Supan 1340 and P7-45FG-0791 to Figure 12. Also, we changed and added the sentences in line 356-359 as below.
Before the change
‘Fig. 12 displays the orientation analysis results for the Supran 1340 materials.’
After the change
‘Fig. 12 displays the fiber orientation and the orientation at skin analysis results for the materials. Figure 12(a) and (b) are the fiber orientation analysis results by direction of Supan 1340 material and P7-45FG-0791 material, and Figure 12(c) is the orientation at skin analysis result of RB84HP material.’
Point 14: What version of Modflow was used?
Response 14: We used moldflow advisor 2018, and we changed ‘moldflow’ in line 118 to ‘moldflow advisor 2018’.
Point 15: What type of analysis was chosen? E.g. Fill+Pack+Cool+Warp?
Response 15: For this simulation Fill+Pack+Cool+Warp Analysis was chosen. And we inserted the following phrase in line 195.
‘And this simulation was chosen the Fill+Pack+Cool+Warp analysis.’
Point 16: It is worth including the pVT curve from the Tait model in the work (it is in the material base of the polymer composite used).
Response 16: Figure 5 shows the addition of the 2-states Tiet model pVT curve, and the contents of lines 171-174 are modified.
Before the change
‘The viscosity-shear rate graph of Supran 1340 material used in the analysis is dis-played in Fig. 5, demonstrating viscous thinning behavior. Similarly, RB84HP and P7-45FG-0791 also exhibit similar trends.’
After the change
‘Figure 5 shows The visibility-shear rate graph and 2-states Tiet model pVT curve of Supran 1340 material. Supran 1340 material shows demonstrating viscous thinning be-havior and is identified as a crystalline resin. Similarly, RB84HP and P7-45FG-0791 also exhibit similar trends.’
Point 17: Figure 12 - is it really a fiber orientation simulation because I'm not sure?.. please check...
Response 17: The results shown in Figure 12 are not the fiber orientation results. They are reuslts for the orientation of the fibers at the skin. The fiber information for one of the materials (RB84HP) used for this study is not provided, therefore it was not possible to generate fiber orientation resluts for this material. Hence we used the results for orientation at skin.
However, following your suggestion in comment point 13, we have added 1st, 2nd, and 3rd order fiber orientation results of the materials whose fiber information were available.
Point 18: What was the shrinkage of the simulation product?
Response 18: The maximum volumetric shrinkage at ejection for SUPRAN1340, RB84HNKOR and P73G45 were found to be 15.53%, 14.85% and 14.14% respectively. And I added the volumetric linkage analysis section in line 375-392 as below
‘After injection molding, Moldflow analysis was performed to predict the volume shrinkage rate of the product. The volumetric shrinkage analysis shows the estimated volumetric shrinkage of the product when the resin is cooled to room temperature. The volumetric shrinkage is determined by the temperature and pressure of each part of the packing pressure and cooling process. In general, the faster the cooling rate, the higher the packing pressure and the longer the packing pressure time, the lower the volume shrinkage. The volumetric shrinkage affects the dimensions, deformation, and sink mark of the product, and if the differential shrinkage is large, the possibility of contraction deformation increases. Table 12 presents the volumetric shrinkage results, and Fig. 13 shows the volumetric shrinkage analysis results for Supran 1340 materials. The maximum volumetric shrinkage showed 15.53% of Supran 1340 materials, 14.86% of RB84HP ma-terials, and 14.14% of P7-45FG-0791 materials. And all three materials have the highest volumetric shrinkage around the gate, as shown in Figure 13, and the volumetric shrinkage has decreased towards the end of the product.’
Table 12. Volumetric shrinkage analysis results.
|
Model |
Estimated max. volumetric shrinkage (%) |
|
Supran 1340 |
15.53 |
|
RB84HP |
14.86 |
|
P7-45FG-0791 |
14.14 |
Figure 13. Volumetric shrinkage analysis results.

Reviewer 3 Report
It is known that the consumption of energy is amazingly high and the fossil based resources may not be able to provide energy for the whole world as these resources will be used up in the near future. Hence, renewable energy expected to play an important role in handling the demand of the energy required along with environmental pollution prevention. Many countries showing great interest towards renewable or green energy generation. The reviewed paper (manuscript ID: processes - 2419048, titled: A Feasibility Study on the Use of Injection Molding Systems for Mass Production of 100W Class Wind Turbine Blades) is very up-to-date and presents results in the field of production of the blades for a spiral-shaped small wind power generator by injection molding. The authors designed and analysed the mold using Moldflow CAE and they also assessed the functionality of the manufactured blade by incorporating it into a spiral small wind generator. I have found the paper to be interesting. I appreciate the effort that the authors have put in performing this study. I have no objections against the work by essence, but I have some comments.
Supplement the introduction with more recent literature.
Suggestion: I would use conformal cooling to cool the mold.
Line 407: What type of universal testing machine was used for tensile and bending tests of materials?
Line 495 : How was the safety and damage checked of the small wind turbine produced?
Author Response
Response to Reviewer 3 Comments
It is known that the consumption of energy is amazingly high and the fossil based resources may not be able to provide energy for the whole world as these resources will be used up in the near future. Hence, renewable energy expected to play an important role in handling the demand of the energy required along with environmental pollution prevention. Many countries showing great interest towards renewable or green energy generation. The reviewed paper (manuscript ID: processes - 2419048, titled: A Feasibility Study on the Use of Injection Molding Systems for Mass Production of 100W Class Wind Turbine Blades) is very up-to-date and presents results in the field of production of the blades for a spiral-shaped small wind power generator by injection molding. The authors designed and analysed the mold using Moldflow CAE and they also assessed the functionality of the manufactured blade by incorporating it into a spiral small wind generator. I have found the paper to be interesting. I appreciate the effort that the authors have put in performing this study. I have no objections against the work by essence, but I have some comments.
Thank you for your good evaluation. We edited the article as below after accepting your advice.
Point 1: Supplement the introduction with more recent literature.
Suggestion: I would use conformal cooling to cool the mold.
Response 1: We inserted the following sentence in line 200-205 and added 4 references.
‘One of the most important factors in injection molding is the cooling channel. The temperature of the mold varies due to the design and conditions of the cooling channel, which has a great influence on injection molding. It affects many factors such as fill time, short shot, ejection time, and deformation of the product, various studies are being con-ducted on cooling channel[21-24]. In this study, cooling channel performance analysis was also conducted to evaluate the performance of the cooling channel.’
- Y. C. Lam; L. Y. Zhai; K. Tai; S. C. Fok; An evolutionary approach for cooling system optimization in plastic injection mould-ing, International Journal of Production Research 2004, 42(10), 2047-2061.
- Muhammad Khan; S. Kamran Afaq; Nizar Ullah Khan; Saboor Ahmad; Cycle Time Reduction in Injection Molding Process by Selection of Robust Cooling Channel Design, ISRN Mechanical Engineering 2014, 1-8.
- Mahesh S Shinde; Kishor M Ashtankar; Additive manufacturing–assisted conformal cooling channels in mold manufacturing processes, Advances in Mechanical Engineering 2017, 9(5), 1-14.
- Mandana Kariminejad; Mohammadreza Kadivar; Marion McAfee; Gerard McGranaghan; David Tormey; Comparison of Conventional and Conformal Cooling Channels in the Production of a Commercial Injection-Moulded Component, Key En-gineering Materials 2022, 926, 1821-1831.
Point 2: Line 407: What type of universal testing machine was used for tensile and bending tests of materials?
Response 2: Tensile and bending tests were carried out using RB 301 UNITECH-M universal testing machine manufactured by R&B company in Republic of Korea. And we inserted the following phrase in line 456-457.
‘Tensile and bending tests were carried out using RB 301 UNITECH-M universal testing machine(R&B Co., Republic of Korea).’
Point 3: Line 495 : How was the safety and damage checked of the small wind turbine produced?
Response 3: The product was visually determined for damage, and the operation was checked by observing the generation of electricity. And we changed the 536 - 539 lines as below.
Before the change
‘Finally, an assembled small wind turbine was installed to observe the operation of the blade, and its safety and damage were checked.’
After the change
‘Finally, an assembled small wind turbine was installed to observe the operation of the blade. The product was visually determined for damage, and the operation was checked by observing the generation of electricity. Therefore, we confirmed that power is produced stably without damage.’

Reviewer 4 Report
Dear Authors,
In general, this manuscript is interesting, but it needs to be refined. It is important above all to highlight the innovativeness of your research. Detailed comments are provided below.
Line 22: Were baskets counted in these studies? If not, then don't mess it up.
Line 68: Each acronym should be explained before first use.
Line 70-72. This fragment is better suited to the research methodology.
In addition, this is where you should better highlight the innovation of your research. The research problem described below seems to be correct.
Line 75: The use of glass fibers as a reinforcing additive for resin biocomposites is well known. What is innovative about this solution? How do you know at this point that the use of these fibers will sufficiently strengthen the windmill's lugs? This is more of a conclusion
Line 83-86: This passage is also methodology and should be placed there.
I think you should better organize this piece of text on lines 68-86.
Line 113: This analysis should be better described. Each software used in research must contain (name: producer, city, country). Review the entire methodology in this respect.
Table 1: Table header is missing. Also, I understand that the values in the table are dimensionless. However, it is good that you mark it somewhere in the text.
Line 415: Write the type of machine on which the strength tests were performed. In addition, the description of these studies should also be included in the methodology.
Table 14: Write down how many strength tests were carried out. Enter the standard deviation, SD, or other statistic.
Line 440: name: manufacturer, city, country. Add such a description. Was it a twin screw injection molding machine? Also, were the screws counter-rotating or co-rotating?
Line 452: Mark on the drawing or describe where (on the cylinder of the injection molding machine) the individual temperatures H1-H5 were set. Whether H1 was the value at the beginning or end of the injection molding machine cylinder.
Line 476: These are not conclusions, just a summary. Conclusions from your research should be well formulated. For example, what does your research contribute to the development of wind turbines? What are the prospects? etc. This is very important.
Line 514: The literature is quite poor, you should add some more items.
Author Response
Response to Reviewer 4 Comments
Dear Authors,
In general, this manuscript is interesting, but it needs to be refined. It is important above all to highlight the innovativeness of your research. Detailed comments are provided below.
Thank you for your professional evaluation. We accepted the advice of Reviewer and revised the article as below, and wrote the answer as below.
Point 1: Line 22: Were baskets counted in these studies? If not, then don't mess it up.
Response 1: We changed ‘The results indicated the feasibility of producing a blade for a small wind turbine through injection molding, which showed higher productivity and lower costs compared to traditional manufacturing methods that rely heavily on manual labor.’ in line 20-23 to ‘The results indicated the feasibility of producing a blade for a small wind turbine through injection molding, which predicted higher productivity and lower costs compared to traditional manufacturing methods that rely heavily on manual labor.’.
Point 2: Line 68: Each acronym should be explained before first use.
Response 2: We changed ‘(CAE)’ in line 68 to ‘(Computer Aided Engineering: CAE)’.
Point 3: Line 70-72: This fragment is better suited to the research methodology.
In addition, this is where you should better highlight the innovation of your research. The research problem described below seems to be correct.
Response 3: As you advised, we highlighted the sentence by mentioning to the research methodology. And we changed the 70 - 73 lines as below.
Before the change
‘Additionally, studies have been conducted to select appropriate materials for injection molding by inputting the characteristics of various feedstock materials into an analysis program prior to the actual molding process, thus reducing trial-and-error efforts [19-20].’
After the change
‘In addition, in terms of research methodology, studies have been conducted to select appropriate materials for injection molding by inputting the characteristics of various feedstock materials into an analysis program prior to the actual molding process, thus reducing trial-and-error efforts [19-20].’
Point 4: Line 75: The use of glass fibers as a reinforcing additive for resin biocomposites is well known. What is innovative about this solution? How do you know at this point that the use of these fibers will sufficiently strengthen the windmill's lugs? This is more of a conclusion.
Response 4: We changed ‘ To address this issue, researchers have determined that the use of glass fibers in combi-nation with thermoplastic resin during the injection molding process can result in blades with sufficient strength.’ in line 75-78 to ‘To address this issue, researchers needed to research whether a combination of thermoplastic resin and glass fibers during injection molding process could create a blade with sufficient strength.’.
Point 5: Line 83-86: This passage is also methodology and should be placed there.
I think you should better organize this piece of text on lines 68-86.
Response 5: Lines 79-87 describe the purpose and method of the study, and lines 75-78 have been changed according to the reviewer's advice.
Point 6: Line 113: This analysis should be better described. Each software used in research must contain (name: producer, city, country). Review the entire methodology in this respect.
Response 6: We changed 'Moldflow' in line 121 to 'Moldflow Advisor 2018' after accepting your advice.
Point 7: Table 1: Table header is missing. Also, I understand that the values in the table are dimensionless. However, it is good that you mark it somewhere in the text.
Response 7: We inserted the table header in Table 1.
Point 8: Line 415: Write the type of machine on which the strength tests were performed. In addition, the description of these studies should also be included in the methodology.
Response 8: Tensile and bending tests were carried out using RB 301 UNITECH-M universal testing machine manufactured by R&B company in Republic of Korea. And we inserted the following phrase in line 456-457.
‘Tensile and bending tests were carried out using RB 301 UNITECH-M universal testing machine(R&B Co., Republic of Korea).’
Tensile and bending tests were conducted in accordance with the American Society for Testing and Materials (ASTM) specification, which is shown in lines 457 - 460.
Point 9: Table 14: Write down how many strength tests were carried out. Enter the standard deviation, SD, or other statistic.
Response 9: The information is in lines 454 - 455, five specimens were tested for each material, and the test results were averaged.
Point 10: Line 440: name: manufacturer, city, country. Add such a description. Was it a twin screw injection molding machine? Also, were the screws counter-rotating or co-rotating?
Response 10: The manufacturer of the injection machine is UBE, and the country of manufacture is Japan. It is single screw injection molding machine, not twin screw injection molding machine.
And also We changed 'the UBEMAX-ST series 1300-280 injection molding machine' in line 481-482 to 'the UBEMAX-ST series 1300-280 injection molding machine(UBE Co., Japan).
Point 11: Line 452: Mark on the drawing or describe where (on the cylinder of the injection molding machine) the individual temperatures H1-H5 were set. Whether H1 was the value at the beginning or end of the injection molding machine cylinder.
Response 11: Table 15 has been modified as follows so that the location of the heater can be intuitively identified.
Before the change
|
Heater number |
Cylinder temperature (℃) |
|
NH |
230 |
|
H1 |
230 |
|
H2 |
230 |
|
H3 |
230 |
|
H4 |
220 |
|
H5 |
210 |
After the change
|
Items |
Numerical values |
|||||
|
Heater number |
NH |
H1 |
H2 |
H3 |
H4 |
H5 |
|
Cylinder temperature (℃) |
230 |
230 |
230 |
230 |
220 |
210 |
Point 12: Line 476: These are not conclusions, just a summary. Conclusions from your research should be well formulated. For example, what does your research contribute to the development of wind turbines? What are the prospects? etc. This is very important.
Response 12: We added the following information to the 540-552 line.
‘Our study is a critical step in the mass production of the product once it is developed. The division of the complex geometry of the spiral blades and the analysis of the material selection method and data will be of great interest to injection molding engineers. Fur-thermore, our work will aid the small wind turbine sector by providing guidelines for mass-producing blades with complex shapes to scholars and engineers studying wind turbines.
It is worth noting that our study specifically focused on the possibility of utilizing injection molding as a means of mass-producing small wind turbine blades. This study is particularly significant because it demonstrates that the injection molding method can be utilized to manufacture blades with complex geometries, which diverges from traditional blade manufacturing techniques. We believe that our findings will contribute to the de-velopment of more efficient and cost-effective manufacturing processes for small wind turbine blades.’
Point 13: Line 514: The literature is quite poor, you should add some more items.
Response 13: We inserted the following sentence in line 200-205 and added 4 references.
‘One of the most important factors in injection molding is the cooling channel. The temperature of the mold varies due to the design and conditions of the cooling channel, which has a great influence on injection molding. It affects many factors such as fill time, short shot, ejection time, and deformation of the product, various studies are being con-ducted on cooling channel[21-24]. In this study, cooling channel performance analysis was also conducted to evaluate the performance of the cooling channel.’
- Y. C. Lam; L. Y. Zhai; K. Tai; S. C. Fok; An evolutionary approach for cooling system optimization in plastic injection mould-ing, International Journal of Production Research 2004, 42(10), 2047-2061.
- Muhammad Khan; S. Kamran Afaq; Nizar Ullah Khan; Saboor Ahmad; Cycle Time Reduction in Injection Molding Process by Selection of Robust Cooling Channel Design, ISRN Mechanical Engineering 2014, 1-8.
- Mahesh S Shinde; Kishor M Ashtankar; Additive manufacturing–assisted conformal cooling channels in mold manufacturing processes, Advances in Mechanical Engineering 2017, 9(5), 1-14.
- Mandana Kariminejad; Mohammadreza Kadivar; Marion McAfee; Gerard McGranaghan; David Tormey; Comparison of Conventional and Conformal Cooling Channels in the Production of a Commercial Injection-Moulded Component, Key En-gineering Materials 2022, 926, 1821-1831.

Reviewer 5 Report
Attached the comments file.

Author Response
Response to Reviewer 5 Comments
The authors present a paper entitled “A Feasibility Study on the Use of Injection Molding Systems for Mass Production of 100W Class Wind Turbine Blades”, in which they have investigated the producing feasibility of a small wind turbine blade through injection molding technique. A complete study of the design of the object, an analysis using Moldflow CAE S/W and a selection of the most suitable material to be injected have been done. An injection-moulded blade was produced and assembled with other components and a generator was installed to assess durability, safety, and performance under various conditions. The considered topic is a current and interesting one. The paper is well-written and well-structured, presenting the flow of the experimental activities in a clear and coherent way. For this reason, I suggest accepting the paper after minor editing revisions.
Thank you for your good evaluation. We edited the article as below after accepting your advice.
Point 1: Line 30: to-wards → towards
Response 1: We changed ‘to-wards’ to ‘towards’.
Point 2: Line 34: in-creased → increased
Response 2: We changed ‘in-creased’ to ‘increased’.
Point 3: Line 132: dis-played → displayed
Response 3: We changed ‘dis-play’ to ‘display’.
Point 4: Line 141: in the caption it is Figure 4 and not Figure 2
Response 4: We changed 'Figure 2' in line 146 to 'Figure 4'.

Round 2
Reviewer 2 Report
The current version of the publication is recommended for acceptance.
Reviewer 4 Report
Dear Athors,
Accepts all corrections.